# Does host-guest interaction promote tolerance behavior? The mediating role of place attachment and subjective well-being

Yajun JIANG[1], Longfang HUANG[1,2]*, Huiling ZHOU[1]*, Ke WU [3]*

1 College of Tourism & Landscape Architecture, Guilin University of Technology, Guilin, Guangxi, China,
2 College of International Tourism and Public Administration, Hainan University, Haikou, Hainan,China,
3 School of Economics & Management, Hunan University of Science and Engineering, Yongzhou, Hunan, China

* emiliafang@163.com (LFH); huiling_zhou@163.com (HLZ); huse_chn@163.com (KW)

## Abstract

This study explores the mediating role of place attachment (PA) and subjective well-being (SWB) in the mechanism of tourist tolerance behavior (TTB) from the perspective of host-guest interaction (HGI). Taking the Two Rivers and Four Lakes Scenic Area and Xingping Ancient Town in Guilin, a world-renowned tourist destination in China, as research cases, this study collected survey questionnaires from tourists in the scenic spots and used structural equation modeling to test the theoretical hypotheses of TTB. The results show that HGI has a significantly positive effect on TTB. Tourists' PA and SWB play a partial mediating role between HGI and TTB. In addition, the study further verified the role of PA and SWB as chain mediators between HGI and TTB. This study not only expands the research scope of tourist citizenship behavior, but also reveals the key factors to improve TTB in tourist destinations. Finally, it provides relevant insights from the perspective of tourist destination management and services.

## Introduction

In recent years, the swift advancement of an open economy and the dissemination of civilized tourism principles have prompted a growing number of tourists to willingly engage in pro-social behaviors that benefit tourism enterprises or destinations, referred to as tourist civic behaviors [1].These behaviors are characterized by their spontaneous and voluntary nature, generating value for stakeholders without being obligatory [2,3].They encompass activities such as providing positive word-of-mouth recommendations [4],offering assistance, giving feedback [5],and demonstrating tolerance [2].Within the value co-creation framework between tourists and service providers [2],tourists' tolerance behavior assumes a pivotal role in fostering the long-term sustainable development of the tourism sector and enhancing its competitive edge

**Data availability statement:** All relevant data are within the manuscript and its Supporting Information files.

**Funding:** This research was supported by the National Natural Science Foundation of China [Grant No. 72064006 to ZH and no. 72462013 to JY]. The funders played a role in study design, data collection and analysis, decision to publish, and preparation of the manuscript.

**Competing interests:** The authors have declared that no competing interests exist.

[5,6].Consequently, tolerance behavior has garnered increasing scholarly attention [2],as it not only aids in balancing the often uneven relationship between customers and service providers [7],but also actively contributes to the enhancement of tourists' travel experiences and overall tourism quality.

However, existing research on tourist tolerance behavior (TTB) predominantly focuses on the hospitality sector [8–11].It is important to highlight that, within the relatively confined environment of hotels, customer interactions primarily occur between guests and hotel personnel. Conversely, tourist destinations present a more open setting where visitors engage not only with destination staff, such as tour guides and park attendants, but also regularly with local inhabitants through unstructured encounters. Consequently, the direct application of tolerance behavior influence mechanisms from the hotel context to tourist destination scenarios is constrained by these differences [8–10].

Next, we delve into the relationship between tourists and tourist destinations, with particular emphasis on the host-guest interaction (HGI), which is widely recognized as a pivotal factor in the sustainable development of tourist destinations [12].The HGI, as a distinctive tourism attraction element, significantly influences tourists' perceptions, attitudes, and behaviors [13].In tourism resources of universal value and substantial significance, such as natural and cultural heritage sites, tourists are inclined to place greater importance on the quality of host-guest interaction [14].Studies indicate that when tourists form a "warm partnership" with local residents, this relationship effectively enhances their engagement [15] and fosters the creation of positive word-of-mouth and sustained visitation intentions [4]. Consequently, in both open natural heritage and cultural heritage tourist destinations, thoroughly investigating how HGI impacts TTB not only contributes to the enrichment of customer behavior theory but also offers novel perspectives and strategies for enhancing tourism service quality.

The literature contribution of this study to TTB is mainly reflected in the following aspects: Firstly, it underscores the critical role of HGI in shaping TTB within tourist destinations. By establishing HGI as a fundamental prerequisite for TTB, this research addresses whether tourists who engage more interactively are indeed more tolerant. Secondly, it elucidates the underlying mechanisms through which HGI influences TTB by incorporating PA and SWB. Lastly, this investigation is situated within the context of open tourist attractions, thereby broadening the scope of research on tolerance behavior in tourism and offering a theoretical framework for exploring analogous relationships in related fields. Consequently, this study paves new avenues for future inquiries into civic behavior among tourists. More broadly, it provides essential theoretical support and practical insights aimed at enhancing TTB in tourist destinations.

## Theoretical basis and hypothesis

### SOR theory

In 1974, Mehrabian and Russell expanded upon the S-O-R theory within the framework of environmental psychology.They posited that various elements of the external environment serve as stimulus factors(S), which influence an individual's cognitive, emotional, and physiological states(O), ultimately shaping the individual's attitudes

and behavioral responses(R) [16]. The organism, representing the internal emotional and cognitive condition of tourists [17], acts as a mediator between external environmental stimuli and tourists' behavioral responses [16].

The validity of the SOR theory has been extensively empirically tested across various domains. Existing research demonstrates that emotional interaction, characterized by familiarity and intimacy as environmental stimuli, plays a pivotal role in influencing users' purchase intention processes [18]. Interactions among users positively impact user perception, facilitating a deeper understanding of the product and enhancing user behavior [19]. In marketing, leveraging social media can augment customers' SWB, thereby fostering increased brand loyalty [20]. Within the tourism sector, positive HGI, destination image, trust, and attitudes positively promote intentions to support tourism [21]. Furthermore, natural empathy and perceived environmental responsibility mediate the relationship between online interactions and consumers' intentions toward low-carbon tourism behaviors [22].

Nonetheless, there remains a paucity of research examining the influence of HGI on TTB. Consequently, this study employs the SOR theory as its theoretical framework. Specifically, HGI serves as the stimulus factor, while place attachment (PA) and SWB function as organism factors reflecting tourists' internal emotional states. TTB is considered the behavioral response. This paper constructs a structural equation model to investigate not only the impact of HGI on TTB but also the mediating role of PA and SWB in this relationship.

## Attachment theory

Attachment theory, which stems from the mother-infant bond theory, is a pivotal framework in the study of human relationships and provides a comprehensive structure for understanding the development of emotional bonds [23,24]. Attachment refers to the inherent human tendency to form emotional connections with specific entities, and this inclination can significantly influence an individual's judgment. Scholars generally agree that the attachment bond between individuals and places can foster a sense of security and well-being [25]. Hong's research demonstrates a positive correlation between an individual's value attachment and SWB [26]. Ujang further highlights that tourists' PA can enhance the generation of well-being [27]. DiWu et al. found that the impact of tourists' PA on SWB varies across different dimensions [28].

Emotion, as a fundamental component of the human-environment relationship [29], occupies a pivotal role in attachment theory. This theory suggests that subjective emotional states significantly influence human behavioral choices [30]. For example, Wang et al. utilized attachment theory to conduct an in-depth analysis of the social attachment factors underlying community members' purchasing behaviors in business activities [31]. Shallcross et al. examined the potential link between attachment orientation and the propensity to share positive news within the framework of attachment theory [32]. Cohn posited that well-being serves as an external manifestation of emotional states and can directly impact behavioral outcomes [33]. Individuals experiencing positive emotions tend to exhibit higher levels of well-being, which in turn positively reinforces behavioral outcomes [33]. Furthermore, research indicates that SWB mediates the relationship between PA and pro-environmental behavior [34]. Consequently, this paper hypothesizes that PA not only influences SWB but also that both PA and SWB may function as mediating variables in the process by which HGI affects TTB.

## HGI and TTB

The essence of interaction is the activity of communication between subjects and objects [35]. HGI refers to personal contact between tourists and hosts [36]. The interactive subjects in tourism include service providers, residents, guides, stakeholders, sales personnel, and so on, who are the initiators and organizers of activities. The object refers to the party receiving the service, such as tourists, customers, visitors, etc. In general, HGI aims to meet the needs of tourists and the tourism industry, rather than meet the needs of local residents for cultural exchanges, so tourists often occupy a central position in this interaction [37]. There are numerous interactive studies in the field of tourism marketing. For consumers, interaction with marketers can actively promote consumers' green purchasing behaviors [38]. For hotel staff, interaction with guests increases the possibility of workplace deviance [39]. In the process of tourism, the HGI strongly promotes the

green behavior of tourists [40] and the influence of tourists' environmentally responsible behaviors [41]. In this paper, the HGI is defined as the specific, face-to-face and direct social interaction between community residents and tourists.

Tourist citizenship behavior is rooted in organizational citizenship behavior and customer citizenship behavior. refers to the behaviors independently decided and voluntarily displayed by tourists, which can directly or indirectly benefit the tourist destination [5]. According to the definition of customer tolerance behavior [42], this paper defines TTB as the extent to which tourists are willing to tolerate travel inconvenience during the travel process. According to the social exchange theory, people will exchange behaviors according to their own interests and expectations in social communication. When they feel that the benefits outweigh the costs, they will be more inclined to continue the communication; otherwise, they will be disgusted and withdraw from the communication [2]. That is to say, when residents offer assistance and services to tourists, tourists will perceive the efforts of local residents, thus producing a sense of psychological feedback and being more willing to tolerate. Studies have shown that HGI promotes residents' active and passive promotion behaviors [43], stimulates tourists' intention to participate in the co-creation of public services [15], and Promoted tourists' environmentally responsible behavior [44]. Based on the aforementioned analysis and SOR model, we believe that HGI as a stimulus factor in this study may trigger TTB in tourist destinations.Therefore, this paper proposes the following hypothesis:

**H1).** HGI has a significant positive effect on TTB.

## The mediating role of PA

The concept of attachment involves a specific inclination or connection towards a particular object [45], signifying an emotionally charged bond between an individual and a particular object [46]. In the field of environmental psychology,PA refers to the psychological connection between an individual and a specific place or environment [47]. It encompasses two interrelated dimensions [48]: place identity, which pertains to the significance of resources for engaging in activities (i.e., functional dependency), and place dependence, which relates to the extent emotions or symbolic meanings are attributed to a particular place (emotional dependency). Therefore, this study considers PA as a one-dimensional latent variable, emphasizing the components of local identity and local dependence [49].

PA is not a static phenomenon; rather, it is dynamic and evolves as tourists engage with their travel destinations, gradually fostering a robust emotional connection [50]. According to the SOR theory, individuals are significantly influenced by external stimuli and environmental factors, resulting in corresponding behavioral responses [51]. As a crucial component of travel experiences, HGI is anticipated to enhance the experiential value for tourists and further reinforce their PA towards the destination [14]. For instance, Woosnam discovered that interactions between tourists and local residents at World Heritage sites contribute to mutual attachment to these locations [52]. Additionally, Wang et al. highlighted that interactions between tourists and service providers positively influence PA levels [53]. Consequently, this study posits that HGI can effectively stimulate and strengthen tourists' attachment to their travel destinations. Based on this premise, we propose the following hypothesis:

**H2).** HGI has a significant positive effect on PA.

Attachment theory posits that PA, defined as a deep emotional bond between individuals and their environments, is a critical psychological construct that substantially influences tourists' behaviors and decision-making processes [54]. Wang et al. utilized attachment theory to elucidate the motivations behind purchasing behaviors among community members in commercial contexts and evaluated the influence of social attachment on such behaviors [31]; Shallcross et al. explored the correlation between attachment orientation and the propensity to share positive news within the framework of attachment theory [32]. Moreover, prior research has demonstrated that PA exerts a positive and significant effect on tourists' environmental responsibility behaviors [55], behavioral intentions [56], and revisit intentions [57]. Building on this body of literature, this study hypothesizes that PA can positively impact TTB. Consequently, the following hypothesis is proposed:

**H3)**. PA has a significant positive effect on TTB.

The SOR theory posits that external environmental stimuli can elicit emotional responses within individuals, which subsequently lead to the development of corresponding attitudes and ultimately influence behavioral responses. PA is a kind of spiritual bond generated during the interaction between tourists and tourist destinations [50]. When individuals have a higher positive emotion towards the tourist destination, they will show a strong behavioral inclinations [54]. Moreover, PA serves as a mediating factor in various relational dynamics, such as those between restorative sensory perception and environmentally responsible behavior [55], as well as gastronomic experiences and behavioral intentions [56]. These findings underscore the viability of PA as a mediator and its substantial impact on individual behavior. Consequently, we hypothesize that PA may evoke specific behavioral responses when stimulated by HGI. Based on this premise, we propose the following hypotheses:

**H4)**. PA plays a mediating role between HGI and TTB.

### The mediating role of SWB

Well-being, as an important aspect of an individual's overall quality of life and satisfaction with life, has become one of the most valuable concepts in social science disciplines [58,59]. From the perspective of psychology, people's subjective well-being(SWB) refers to individuals' feelings and evaluation over their own life status, which varies from person to person [60]. In essence, SWB seeks to understand what makes people happy and satisfied with their lives [61]. In this study, the SWB of tourists is defined as their feelings and satisfaction with the tourist destinations and travel experiences.

Positive interaction is an important driver of tourists' SWB [62]. A lack of social interaction can lead to a diminished sense of well-being [63]. Conversely, even minimal social engagements can significantly enhance SWB in everyday life [64], contributing to greater happiness among individuals [65,66].Existing literature indicates that teacher-student interactions positively and significantly influence students' classroom SWB [67],HGI on residents' happiness [68,69], and human interaction on tourists' SWB [70]. Moreover, HGI offers tourists diverse travel experiences [15], which in turn can affect their SWB [71]. Therefore, the interaction between tourists and residents may have an impact on the SWB of tourists. Based on this premise, we propose the following hypothesis:

**H5).** HGI has a significant positive effect on SWB.

The theory of emotion regulation in psychology posits that an individual's emotional state and experiences can significantly influence their behavior [72]. For instance, as women's sense of happiness increases, they may exhibit greater resilience to the pressures associated with economic and caregiving responsibilities [73]. Likewise, the SWB of tourists during their travels has a positive and substantial effect on their decision-making behaviors regarding travel [74], intentions to revisit [75], and overall loyalty [76]. Drawing from prior research, this study asserts that SWB is a critical factor affecting tourists' behavioral intentions. Contented tourists are more inclined to demonstrate tolerance in their actions. Consequently, we propose the following hypothesis:

**H6).** SWB has a significant positive effect on TTB.

The SOR theory posits that external environmental stimuli can elicit emotional responses within the body, which subsequently influence behavioral reactions. Moreover, SWB serves as a mediator in various relational dynamics, such as the connection between landscape appeal and intentions for repeat visits [75], as well as the association between nostalgic emotions and intentions for repeat visits [77]. These findings underscore the viability of SWB as a mediating factor and its substantial impact on individual behavior. Consequently, we propose that SWB, representing an individual's most authentic feelings, can be conceptualized as an organism's subjective experience. In response to HGI stimulation, this subjective experience may provoke corresponding behavioral reactions. Building upon this premise, we further advance the following hypothesis:

**H7).** SWB plays a mediating role between HGI and TTB.

**The chain mediating effect of PA and SWB**

Attachment theory is not only pertinent to interpersonal relationships [78], but it also serves as a framework for elucidating the emotional connection between individuals and their environments [79,80]. Empirical evidence demonstrates that tourists can develop a distinctive emotional attachment to their travel destinations, which significantly influences their SWB [27,28]. More specifically, the emotional bond formed between an individual and a particular region can provide psychological security and a sense of social belonging [30], thereby effectively enhancing SWB [81,82]. Furthermore, research indicates that PA is linked to both physical and mental health [83,84] and can foster positive emotional and behavioral responses [85,86]. Based on these findings, this study proposes the following hypothesis:

**H8).** PA has a significant positive effect on SWB.

In prior research, PA and SWB have been extensively validated as critical mediating factors. Specifically, PA mediates the relationship between local social identity and well-being [87], while community attachment serves as a mediator between community environmental perception and residents' SWB [88]. Furthermore, Lin highlighted that SWB functions as an intermediary between PA and pro-environmental behavior [34]. Wang demonstrated that PA and SWB exert a sequential mediating effect on the influence of environmental cognition on the loyalty of rural homestay summer vacationers [89]. Drawing on SOR theory, attachment theory, and the aforementioned hypotheses, we posit that HGI, as an external stimulus, can influence TTB by enhancing their PA and SWB. Based on these theoretical frameworks and empirical findings, this study proposes the following hypothesis:

**H9).** PA and SWB play a chain-mediated role between HGI and TTB.

In summary, a model diagram of the influence of the interaction between HGI on TTB is proposed, shown as Fig 1.

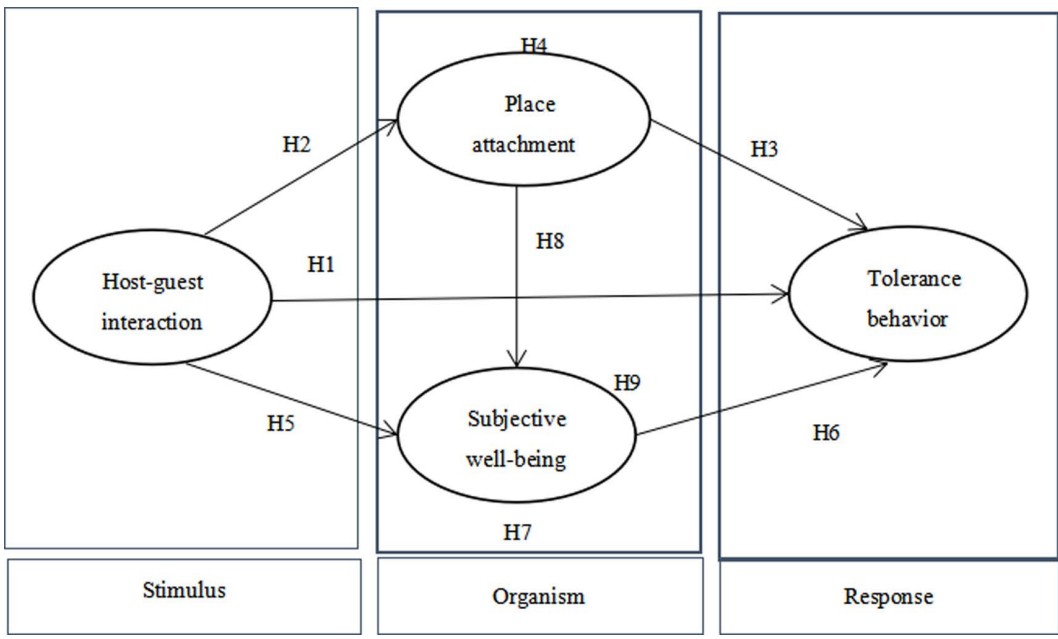

**Fig 1. Hypothetical model.**

## Methodology

### Study site

Guilin, renowned as one of China's premier tourist destinations, holds a distinguished place among the first batch of National Famous Historical and Cultural Cities. It is celebrated for its unparalleled natural beauty, often lauded with the phrase "Guilin's landscape is the finest under heaven". A notable highlight is the Two Rivers and Four Lakes Scenic Area, a 5A-rated attraction that falls under the category of natural heritage sites. This scenic belt encircles the city center of Guilin, presenting a captivating ring-shaped landscape that integrates the Li River, Peach Blossom River, Banyan Lake, Fir Lake, Laurel Lake, and Mulong Lake into a harmonious urban tableau. Another noteworthy site is the Xingping Ancient Town Scenic Area, a 4A-rated attraction categorized as a cultural heritage site. Known as "the luminous pearl on the Li River", Xingping Ancient Town is situated in Yangshuo County, Guilin City, and boasts over a millennium of history. The town is replete with Ming and Qing Dynasty architecture and invaluable cultural artifacts, offering visitors an opportunity to delve into its rich historical and cultural tapestry.

The selection of the Two Rivers and Four Lakes Scenic Area and Xingping Ancient Town in Guilin City as research sites is justified by several reasons. Firstly, these two attractions offer rich tourism resources with distinct characteristics. The former belongs to the natural heritage category of scenic spots, while the latter is a cultural heritage category of scenic spots. Secondly, both attractions are open scenic spots, that is, there are no walls, no clear boundaries, and there are local residents living and working inside, tourists can freely explore the landscape and culture, which is more likely to lead to HGI. Finally, gathering samples from two distinct categories of tourist attractions and performing independent data analyses contributes to enhancing the validity and robustness of our research outcomes.

### Measurement tool

This study has received approval from the Academic Committee of the College of Tourism and Landscape Architecture of Guilin University of Technology, under approval number TLAC20221027001. Prior to commencing the study, we provided comprehensive explanations regarding its content to all participants and underscored that it did not involve any personal privacy concerns. Consequently, I obtained verbal consent from participants to complete the questionnaire. Although a written informed consent form was not utilized, I implemented additional measures during the investigation, which included photographing and documenting the consent process of participants—capturing images of individuals explicitly expressing their willingness to participate in the study as well as clear verbal and/or nonverbal affirmations (e.g., nodding).

The variables were measured with adopted well-established scales modified as appropriate for the background of the survey. All measurements used a five-point Likert scale, with "1" being strongly disagree and "5" being strongly agree. The specific sources of items measured for each variable are as follows.host-guest interaction (HGI), adapted from the scale of Stylidis [90], with 4 items; place attachment (PA), adapted from Williams et al's [91] scale, with 6 items; subjective well-being (SWB), adapted from Wang et al.'s [92] scale, with 4 items; tourist tolerance behavior (TTB), adapted from Yi et al.'s(2) scale, with 4 items. Additionally, the questionnaire included questions on demographic characteristics (gender, age, education, monthly income, etc.).

### Pre-test of the measures

In order to study the influence of HGI on TTB and its potential mechanism, the existing maturity scale was adapted in combination with the research. The specific items are shown in Table 1. Three experienced tourism experts were invited to evaluate the validity of the project content and make suggestions on the sentence and structure of the questionnaire. The updated questionnaire was then provided to 60 undergraduate hospitality/tourism students [93] to initially assess the reliability and validity of the survey tool. The results show that the KMO value of each variable is greater than 0.7. The pre-test results show that the scale is reliable and effective. The Amos22.0 two-step approach was adopted in all the studies, first estimating the measurement model and then analyzing the structural path model to test the theoretical model.

**Table 1. Exploratory factor analysis (pilot study).**

| Variable | Dimension | Factor1 | Factor2 | Factor3 | Factor4 |
|---|---|---|---|---|---|
| HGI | Local residents will explain the local way of life to me | 0.843 | | | |
| | I enjoy building friendships with local residents | 0.803 | | | |
| | Local residents will recommend me good food and places to visit | 0.771 | | | |
| | Local residents are willing to help me when I need help | 0.799 | | | |
| PA | This place has special significance for me | | 0.860 | | |
| | This is more in line with my travel expectations than any other places | | 0.781 | | |
| | This is a very special place for me | | 0.771 | | |
| | I am very attached to this place | | 0.805 | | |
| | A trip to this destination helps me identify myself | | 0.798 | | |
| | I have a strong identification with the destination | | 0.789 | | |
| SWB | This place fulfills my overall need for well-being | | | 0.865 | |
| | This place plays a very important role in my well-being | | | 0.853 | |
| | This place plays an important role in my travel well-being | | | 0.840 | |
| | This place plays an important role in improving my quality of life | | | 0.863 | |
| TTB | I am willing to be patient if the staff of the destination make a mistake in providing the tour service | | | | 0.78 |
| | I am willing to accept if I have to wait for a longer time in the process of receiving travel services | | | | 0.803 |
| | I can understand that prices in tourist areas are moderately high | | | | 0.800 |
| | I can understand if the attractions or reception facilities are not as good as expected | | | | 0.801 |

## Study 1: empirical analysis and findings

### Data collection

Data for Study 1 were gathered between November 24 and December 8, 2022, employing a convenience sampling technique to disseminate paper-based questionnaires among visitors to the Two Rivers and Four Lakes Scenic Area. To enhance the representativeness of the sample, researchers strategically approached tourists across various locations and time periods within the scenic area, aiming to encompass a broader spectrum of visitor demographics. Prior to questionnaire distribution, researchers conducted preliminary screening inquiries to ascertain participants' status as tourists. Subsequently, a concise overview of the study's objectives and questionnaire contents was provided to potential respondents, followed by a request for their voluntary participation. Questionnaires were only distributed upon receiving an explicit indication of willingness to partake in the survey. A total of 353 respondents participated in the questionnaire survey, and 353 questionnaires were recovered, of which 30 were incomplete and the rest 323 were left for analysis, with an effective rate of 91.5%. The characteristics of respondents in study 1 are shown in Table 2.

### Common method bias

To mitigate the influence of common method bias, this study implemented pre-test control measures, including randomizing the order of items, and employed the Harman single-factor method to assess the presence of common method bias within the data. The findings indicated that measurement items were consolidated into four distinct factors. Notably, the first factor accounted for only 22.612% of the total variance across all items, falling below the critical threshold of 40%, thereby suggesting that effective controls were in place to address potential common method variance in this study.

### Measurement model

By analysing the data with Amos software, the measurement model of study 1 was well fitted (see Table 3).Cronbach's alpha values ranged from 0.830 to 0.938, all greater than 0.7, meeting the criteria [94]. Factor loadings for all items (see Table 4) were

**Table 2. Demographic characteristics of respondents.**

| Characteristics | | Study 1: Two Rivers and Four Lakes Scenic Area | | Study 2: Xingping Ancient town | | Study 3: Overall sample | |
|---|---|---|---|---|---|---|---|
| | | Frequency | % | Frequency | % | Frequency | % |
| **Sex** | male | 152 | 47.1 | 210 | 65 | 362 | 56 |
| | female | 171 | 52.9 | 113 | 35 | 284 | 44 |
| **Age** | 18-30 years old | 170 | 52.6 | 162 | 50.2 | 332 | 51.4 |
| | 31-45 years old | 99 | 30.7 | 111 | 34.4 | 210 | 32.5 |
| | 46-60 years old | 41 | 12.7 | 38 | 11.8 | 79 | 12.2 |
| | Over 60 years old | 13 | 4 | 12 | 3.7 | 25 | 3.9 |
| **Educational background** | High school and below | 58 | 18 | 26 | 8 | 84 | 13 |
| | Junior college | 63 | 19.5 | 110 | 34.1 | 173 | 26.8 |
| | Undergraduate | 152 | 47.1 | 167 | 51.7 | 319 | 49.4 |
| | Postgraduate and above | 50 | 15.5 | 20 | 6.2 | 70 | 10.8 |
| **Monthly income** | Less than 3000 yuan | 95 | 29.4 | 94 | 29.1 | 189 | 29.3 |
| | 3001-5000 yuan | 74 | 22.9 | 116 | 35.9 | 190 | 29.4 |
| | 5001-8000 yuan | 99 | 30.7 | 89 | 27.6 | 188 | 29.1 |
| | More than 8001 yuan | 55 | 17 | 24 | 7.4 | 79 | 12.2 |
| **Occupation** | Student | 101 | 31.3 | 77 | 23.8 | 178 | 27.6 |
| | Corporate personnel | 64 | 19.8 | 56 | 17.3 | 120 | 18.6 |
| | Freelancer | 45 | 13.9 | 35 | 10.8 | 80 | 12.4 |
| | Public officials or public institution personnel | 74 | 22.9 | 64 | 19.8 | 138 | 21.4 |
| | Self-employed people | 27 | 8.4 | 48 | 14.9 | 75 | 11.6 |
| | Retiree | 8 | 2.5 | 35 | 10.8 | 43 | 6.7 |
| | Other | 4 | 1.2 | 8 | 2.5 | 12 | 1.9 |

**Table 3. Fitting index of models.**

| Model | χ2/df | RMSEA | GFI | NFI | RFI | IFI | TIL | CFI |
|---|---|---|---|---|---|---|---|---|
| **Study 1** | | | | | | | | |
| **Structural model** | 1.742 | 0.048 | 0.928 | 0.941 | 0.93 | 0.974 | 0.969 | 0.974 |
| **Measurement model** | 2.237 | 0.062 | 0.915 | 0.925 | 0.91 | 0.957 | 0.948 | 0.957 |
| **Study 2** | | | | | | | | |
| **Structural model** | 1.191 | 0.024 | 0.950 | 0.960 | 0.952 | 0.993 | 0.992 | 0.993 |
| **Measurement model** | 2.182 | 0.061 | 0.911 | 0.926 | 0.913 | 0.958 | 0.951 | 0.958 |

Note: RMSEA = root mean square error of approximation, GFI = goodness-of-fit index, CFI = comparative fit index, NFI = normed fit index, RFI = relative fit index, IFI = incremental fit index, TLI = Tucker-Lewis index, CFI = comparative fit index.

greater than 0.5, which is statistically significant (p<0.001) and meets the criterion [94]. The composite reliability (CR) values ranged from 0.827 to 0.938, with each variable's CR exceeding the minimum threshold of 0.7 as recommended by Hair et al. [95]. Additionally, the average variance extracted (AVE) for all constructs ranged from 0.549 to 0.791, satisfying the established criteria [94]. Therefore, in terms of the overall data, the samples in Study 1 all have high reliability, validity and internal consistency.

## Structural model

According to the recommended criteria [96], the structural model fits the data well (Table 3). HGI has a significant positive effect on TTB ($\beta$ =0.252, $p$ <0.01), which provides support for H1. HGI also has a significantly positive effect on PA

**Table 4. Measure model for study 1.**

| Variable | Dimension | Mean | SD | Estimate | CR | AVE | Cronbach's Alpha |
|---|---|---|---|---|---|---|---|
| HGI | Local residents will explain the local way of life to me | 3.72 | 0.924 | 0.843 | 0.859 | 0.605 | 0.856 |
| | I enjoy building friendships with local residents | 3.51 | 1.05 | 0.736 | | | |
| | Local residents will recommend me good food and places to visit | 3.78 | 0.983 | 0.763 | | | |
| | Local residents are willing to help me when I need help | 3.75 | 0.978 | 0.764 | | | |
| PA | This place has special significance for me | 3.81 | 0.967 | 0.862 | 0.906 | 0.616 | 0.904 |
| | This is more in line with my travel expectations than any other places | 3.76 | 1.016 | 0.757 | | | |
| | This is a very special place for me | 3.87 | 0.912 | 0.776 | | | |
| | I am very attached to this place | 3.71 | 1.047 | 0.796 | | | |
| | A trip to this destination helps me identify myself | 3.65 | 1.045 | 0.738 | | | |
| | I have a strong identification with the destination | 3.73 | 1.026 | 0.773 | | | |
| SWB | This place fulfills my overall need for well-being | 3.64 | 1.167 | 0.911 | 0.938 | 0.791 | 0.938 |
| | This place plays a very important role in my well-being | 3.69 | 1.186 | 0.882 | | | |
| | This place plays an important role in my travel well-being | 3.52 | 1.167 | 0.871 | | | |
| | This place plays an important role in improving my quality of life | 3.56 | 1.218 | 0.894 | | | |
| TTB | I am willing to be patient if the staff of the destination make a mistake in providing the tour service | 3.7 | 1.012 | 0.893 | 0.827 | 0.549 | 0.829 |
| | I am willing to accept if I have to wait for a longer time in the process of receiving travel services | 3.39 | 1.032 | 0.717 | | | |
| | I can understand that prices in tourist areas are moderately high | 3.43 | 1.102 | 0.729 | | | |
| | I can understand if the attractions or reception facilities are not as good as expected | 3.41 | 1.075 | 0.596 | | | |

Note: SD = Standard Deviation; CR = Composite Reliability; AVE = Average Variance Extracted.

($\beta = 0.574$, $p0.001$) and SWB ($\beta = 0.504$, $p0.001$), providing a support for H2 and H5. In addition, PA also has a significant positive effect on tourists' SWB ($\beta = 0.364$, $p < 0.001$) and TTB ($\beta = 0.414$, $p0.001$), which provides a support for H3 and H8. Finally, tourists' SWB also has a significant positive effect on TTB ($\beta = 0.146$, $p0.01$), which provides a support for H6. Fig 2 illustrates the structural model findings.

## Mediating effect analysis

In order to better understand the relationship between HGI and TTB, we conducted mediation effect and chain mediation effect analysis with PA and SWB being as mediators to examine the mediation pathway in the proposed model (Table 5). The theoretical model points out three mediating paths of the relationship between HGI and TTB through PA and SWB. Specifically, HGI→PA→SWB→TTB showed a significant positive effect ($\beta = 0.016$, S.E. = 0.010, $p0.001$); HGI→PA→TTB showed a significant positive effect ($\beta = 0.162$, S.E. = 0.036, $p0.001$); HGI→SWB→TTB also had a significant positive effect ($\beta = 0.038$, S.E. = 0.021, $p0.001$). While controlling for these mediating effects, there was a significant positive effect of HGI on TTB ($\beta = 0.372$, S.E. = 0.054, $p0.001$). These results suggest that PA and SWB play a partially mediating role in the relationship between HGI and TTB, providing support for H4、H7 and H9. In total, 26.5% of variance in TTB was explained by HGI, PA and SWB ($R^2 = 0.265$); 24.9% of variance in SWB was explained by HGI and PA ($R^2 = 0.249$); 21.5% of variance in PA was explained by HGI ($R^2 = 0.215$).

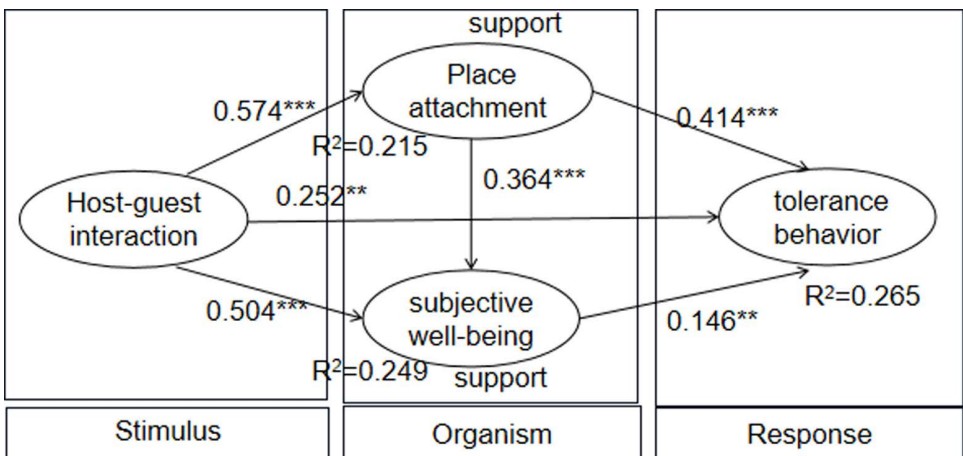

**Fig 2. Structural model results for study 1.**

**Table 5. Mediating effect analysis.**

| Model path | Beta estimate | Standard error | Results |
|---|---|---|---|
| **Study 1** | | | |
| HGI→PA→SWB→TTB | 0.016*** | 0.01 | support |
| HGI→PA→TTB | 0.162*** | 0.036 | support |
| HGI→SWB→TTB | 0.038*** | 0.021 | support |
| HGI→TTB | 0.372*** | 0.054 | support |
| **Study 2** | | | |
| HGI→PA→SWB→TTB | 0.025*** | 0.01 | support |
| HGI→PA→TTB | 0.079*** | 0.03 | support |
| HGI→SWB→TTB | 0.046*** | 0.02 | support |
| HGI→TTB | 0.347*** | 0.053 | support |

Note:

**$p < 0.01$,

***$p < 0.001$.

## Study 2

### Data collection

To examine the proposed model's stability across destinations, a study was conducted among tourists visiting Xingping Ancient Town in Guilin, China, from February 24 to March 10, 2023. Paper questionnaires were administered to tourists within the Xingping Ancient Town scenic area through convenience sampling. To ensure a diverse sample, researchers approached tourists in different sections of the scenic area, at various exits, and during different time periods, thereby encompassing a broader spectrum of tourist demographics. Prior to distributing the questionnaires, researchers conducted preliminary screening to verify the participants' status as tourists. Subsequently, the researchers provided a concise overview of the study's objectives and the questionnaire's content, and sought the consent of potential participants before proceeding with the distribution of the questionnaires. A total of 360 questionnaires were distributed, and 353 questionnaires were collected by the researchers, of which 323 were fully replied. The characteristics of respondents are shown in Table 2.

## Common method bias

To mitigate the influence of common method bias, this study implemented pre-test control measures, including randomizing the order of items, and employed the Harman single-factor method to assess the presence of common method bias within the data. The findings indicated that the measurement items were consolidated into four distinct factors. Notably, the first factor accounted for only 24.125% of the total variance among all items, which falls below the critical threshold of 40%, thereby suggesting that effective controls were in place to address potential common method variance in this study.

## Measurement model

The measurement model of Study 2 was well fitted (see Table 3), Cronbach's alpha values ranging from 0.864 to 0.924, all exceeding the threshold of 0.7, thus satisfying the reliability criteria. Factor loadings for all items, as presented in Table 6, were greater than 0.5 and achieved statistical significance ($p < 0.001$). The composite reliability (CR) of each variable ranged from 0.866 to 0.918, surpassing the 0.7 benchmark, while the average variance extracted (AVE) for all constructs ranged from 0.618 to 0.807, well above the minimum requirement of 0.5 [94]. Therefore, from perspective of the overall data, the samples in Study 2 all have high reliability, validity and internal consistency.

## Structural model

The fitting indices are shown in Table 3. The fitting indices of the structural model are all acceptable [96]. HGI has a significant positive effect on TTB ($\beta = 0.283$, $p0.001$), which provides a support for H1. HGI also has a significant positive effect on PA ($\beta = 0.584$, $p0.001$) and SWB ($\beta = 0.293, p0.001$), providing a support for H2 and H5. In addition, PA also has a significant positive effect on tourists' SWB ($\beta = 0.271$, $p0.001$) and TTB ($\beta = 0.180$, $p0.01$), providing supports for H3 and H8. Finally, tourists' SWB also has a significant positive effect on TTB ($\beta = 0.240$, $p0.001$), providing a support for H6. The fitting indices are shown in Fig 3.

## Mediating effect analysis

In order to better understand the influence of HGI on TTB, the mediation effect and chain mediation effect were examined using the method of Study 1 (see Table 5). The theoretical model points out three mediating paths of PA and SWB between HGI and TTB. Specifically, HGI→PA→SWB→TTB showed a significant positive effect ($\beta = 0.025$, S.E. = 0.010, $p0.001$); HGI→PA→TTB showed a significantly positive effect ($\beta = 0.079$, S.E. = 0.030, $p0.001$); HGI→SWB→TTB also had a significant positive effect ($\beta = 0.046$, S.E. = 0.020, $p0.001$). While controlling for these mediating effects, there was a significant direct positive effect of HGI on TTB ($\beta = 0.347$, S.E. = 0.053, $p0.001$). These results suggest that PA and SWB play a partially mediating role in the relationship between HGI and TTB, providing supports for H4、H7 and H9. In total, 19.4% of variance in TTB was explained by HGI, PA and SWB ($R^2 = 0.194$); 22.9% of variance in SWB was explained by HGI and PA ($R^2 = 0.229$); 18.8% of variance in PA was explained by HGI ($R^2 = 0.188$). The findings of Study 2 further support the results of Study 1.

## Discussion

This study selected tourists from two distinct tourist destinations as research subjects to explore the intrinsic relationships between HGI, PA, SWB, and TTB through the construction of a conceptual model. By systematically collecting and analyzing feedback from tourists at these destinations, this study has not only enriched the body of knowledge on tolerance behavior but also expanded the existing research scope.

Firstly, we have confirmed that HGI exerts a significant positive influence on TTB. The HGI is considered a key driver in promoting the long-term sustainable development of tourist destinations [12]. Prior research has predominantly examined HGI as a prerequisite or moderating variable influencing residents' behaviors [43], with a focus on its effects on tourists'

**Table 6. Measure model for study 2.**

| Variable | Dimension | Mean | SD | Estimate | CR | AVE | Cronbach's Alpha |
|---|---|---|---|---|---|---|---|
| HGI | Local residents will explain the local way of life to me | 3.47 | 1.098 | 0.918 | 0.884 | 0.657 | 0.882 |
| | I enjoy building friendships with local residents | 3.4 | 1.088 | 0.776 | | | |
| | Local residents will recommend me good food and places to visit | 3.42 | 1.085 | 0.786 | | | |
| | Local residents are willing to help me when I need help | 3.48 | 1.099 | 0.751 | | | |
| PA | This place has special significance for me | 3.48 | 1.121 | 0.916 | 0.918 | 0.652 | 0.924 |
| | This is more in line with my travel expectations than any other places | 3.46 | 1.086 | 0.8 | | | |
| | This is a very special place for me | 3.54 | 1.092 | 0.772 | | | |
| | I am very attached to this place | 3.56 | 1.125 | 0.83 | | | |
| | A trip to this destination helps me identify myself | 3.42 | 1.172 | 0.825 | | | |
| | I have a strong identification with the destination | 3.49 | 1.007 | 0.776 | | | |
| SWB | This place fulfills my overall need for well-being | 3.71 | 1.11 | 0.914 | 0.895 | 0.807 | 0.893 |
| | This place plays a very important role in my well-being | 3.7 | 1.129 | 0.818 | | | |
| | This place plays an important role in my travel well-being | 3.74 | 1.055 | 0.785 | | | |
| | This place plays an important role in improving my quality of life | 3.66 | 1.126 | 0.776 | | | |
| TTB | I am willing to be patient if the staff of the destination make a mistake in providing the tour service | 3.69 | 1.088 | 0.885 | 0.866 | 0.618 | 0.864 |
| | I am willing to accept if I have to wait for a longer time in the process of receiving travel services | 3.57 | 1.155 | 0.752 | | | |
| | I can understand that prices in tourist areas are moderately high | 3.59 | 1.153 | 0.717 | | | |
| | I can understand if the attractions or reception facilities are not as good as expected | 3.59 | 1.131 | 0.781 | | | |

Note: SD = Standard Deviation; CR = Composite Reliability; AVE = Average Variance Extracted.

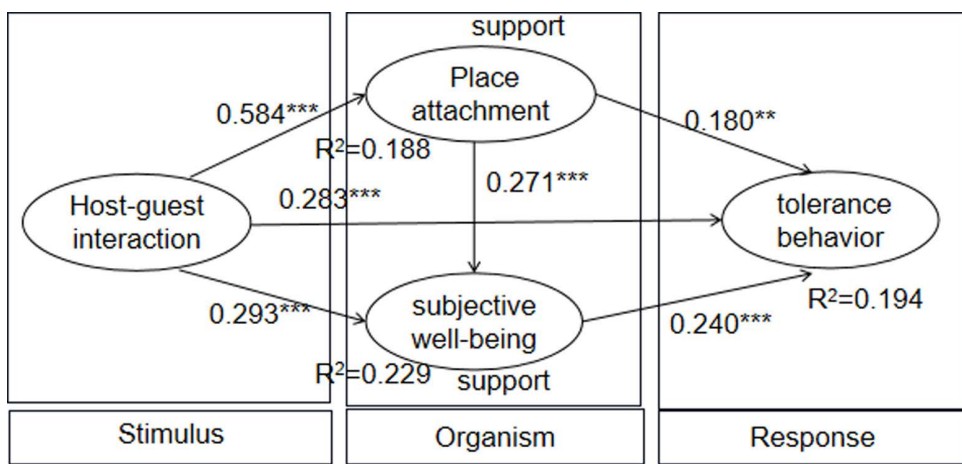

**Fig 3. Structural model results for study 2.**

green behaviors [40] and environmentally responsible actions [41]. Nevertheless, the mechanisms through which HGI impacts TTB are more intricate and multifaceted, and existing research remains inadequate. Tolerant behavior not only facilitates tourists' internal harmony but also offers valuable opportunities for service providers to rectify damaged customer relationships resulting from service failures [97].

Secondly, our research indicates that PA serves as a mediating factor between HGI TTB. PA signifies the evolving emotional connection between tourists and tourist destinations [50], while HGI enhances this attachment. This finding aligns with prior studies demonstrating that HGI significantly influence tourists' cognitive and emotional responses [52]. Furthermore, the strengthening of the human-place emotional bond positively impacts TTB, corroborating existing research that tourists' emotional attachment and sense of identity shape their attitudes toward tourist destinations, thereby influencing their behaviors and decisions [54]. We propose that in the context of open tourist destinations, where tourists have easy access to communication with local residents, this interaction facilitates the reinforcement of the emotional bond between tourists and destinations, leading to greater willingness to tolerate imperfections or negative aspects of the tourist experience.

Thirdly, we have confirmed that SWB acts as a partial mediator between HGI and TTB. SWB represents individuals' cognitive assessments of their own life satisfaction [60]. Prior research predominantly concentrated on the influence of HGI on residents' well-being [68,69], whereas this study broadens the scope to include tourists at tourism destinations, offering a novel perspective on the positive effects of HGI on tourists. Our findings suggest that within the tourism context, HGI can enhance tourists' SWB, and when tourists experience elevated levels of well-being at their destinations, they are more inclined to exhibit understanding and tolerance. This aligns with the theory of emotion regulation, which posits that an individual's emotional state substantially impacts their behavior [72]. Specifically, the communication and interaction between tourists and local residents at tourism destinations not only influence the quality of tourists' travel experiences but also profoundly affect their SWB. Tourists with higher SWB levels are more likely to adopt a positive attitude towards tourism-related unfavorable situations.

Finally, in our study across two tourism destinations, we discovered a chained mediation effect between PA and SWB. By integrating the SOR model with the HGI context, we view PA and SWB as individuals' emotional responses to specific environmental stimuli, and these responses further shape their tolerant behavior. According to emotional cognition theory, individuals who develop a deep emotional attachment to a place are more likely to experience higher levels of SWB. This elevated SWB, in turn, may foster a more open and forgiving attitude, enhancing their ability to positively adapt to different environmental demands and thereby increasing tourists' tolerance for unfamiliar cultures and environments.

## Theoretical implications

This study makes several contributions. Firstly, by investigating the relationships among tourists' HGI, PA, SWB, and TTB, it not only broadens the scope of TTB research and its antecedents but also provides valuable insights for destination managers aiming to enhance destination development. While previous literature on tolerance behavior has predominantly centered on hotel environments [8–10], it has not extensively explored the causal relationship between open-space tourists' HGI and TTB through multiple case studies. It is important to note that open-space tourism destinations differ significantly from closed-space hotels, where customer expectations are primarily focused on highly personalized services, and their tolerance is mainly directed towards minor service errors or shortcomings. In contrast, open-space tourists seek to experience natural beauty, cultural heritage, or entertainment activities, and their tolerance is more often related to inconveniences encountered during their tourism experiences. Considering the positive influence of tourists on destination development [5,6], this study innovatively extends the application of TTB research by validating the impact of open-space tourists' HGI on TTB.

Secondly, this study selected two distinct types of tourist destinations for data collection: the Two Rivers and Four Lakes, which are natural heritage sites, and Xingping Ancient Town, a cultural heritage site. This selection not only encompasses the diversity of tourist destinations but also offers a valuable empirical foundation for investigating the similarities

and differences in visitor behavior patterns across different types of tourist destinations. The data analysis from these two case study sites underscores the broad applicability and robustness of the research findings. It enhances the credibility of the study and provides a solid foundation for future research to promote and apply relevant theories in various types of tourist destinations.

Finally, prior research has emphasized the necessity of a comprehensive investigation into the formation mechanisms of TTB in tourist destinations, particularly focusing on the examination of diverse influencing factors [8–11]. This study adopts the Attitude-Behavior Theory and applies it to the analysis of tolerance behaviors, thereby validating the sequential transmission from PA to SWB, which enhances our understanding of TTB in tourist settings. Furthermore, earlier scholars have recommended conducting additional empirical research to delve into the determinants of PA and SWB among tourists in tourist destinations [14,71]. Our findings indicate that HGI serves as a significant catalyst for PA and SWB, contributing novel insights to the existing body of knowledge on tourist destinations.

## Managerial implications

On one hand, our research suggests that HGI possesses considerable potential to foster tourists' tolerance behavior in tourist destinations. In light of this, destination managers can implement identity incentive strategies aimed at enhancing residents' sense of in-group identity, thereby further encouraging their willingness and actions towards friendliness and participation in HGI. For instance, organizing community events, creating additional job opportunities, and establishing awards such as 'Friendly Residents' or "Outstanding Tourism Service Providers" to acknowledge residents who exhibit a welcoming demeanor and offer thoughtful services to tourists are effective measures. Furthermore, publicity and education play an equally vital role. By thoroughly showcasing the local scenery, customs, cuisine, and entertainment through TikTok, WeChat public accounts and other new media platforms, destination managers can cultivate a deeper understanding among residents regarding their hometowns—thereby bolstering their sense of pride and accomplishment associated with it. This enhanced sense of achievement will motivate residents to engage more actively in amicable HGI while providing sincere warmth in their reception of tourists. Concurrently, we must not overlook the contributions of other tourism stakeholders—particularly tourism operators. Through daily operations and management practices, these operators can gain insights into tourists' needs by incorporating appropriate interactive elements. While adhering to standardized services, they should also flexibly provide personalized offerings tailored to meet specific tourist requirements—effectively addressing emotional needs related to travel experiences at destinations. Such tailored services significantly enhance tourist satisfaction while inspiring them to demonstrate greater social responsibility and civilized conduct; thus contributing positively toward the sustainable development of tourist destinations.

On the other hand, our research underscores the pivotal role of tourists' emotional attachment to the destination and SWB in fostering their tolerance behaviors. This clearly suggests that tourist destinations should prioritize enhancing tourists' PA and SWB to facilitate interactions between visitors and local residents. Destination managers ought to focus on providing humanistic care while fully engaging tourists' PA-SWB and overall happiness. For instance, establishing tourism service stations at strategic locations such as high-speed railway stations, airports, major transfer hubs, and prominent attractions can offer travelers convenient transportation guidance, order maintenance services, complimentary hot water, tourism consultations, among other amenities. These initiatives not only significantly enhance the travel experience for tourists but also expedite their identification with and integration into the destination, thereby nurturing emotional bonds. Furthermore, to strengthen these emotional connections with the destination and elevate tourist happiness levels, destination managers could proactively organize a series of vibrant cultural activities—such as themed music concerts related to the destination, folk performances showcasing local customs, and regional sports events. Such activities are designed to immerse tourists in the local cultural milieu which deepens their attachment and sense of identity with the area. The establishment of this emotional connection will further encourage tolerant behavior among tourists while laying a robust foundation for sustainable development within tourist destinations.

 

## Limitations and future research directions

Although this study has novel implications, as with any other study, some limitations should be further addressed in the future. Firstly, this paper only examined the Chinese tourist population. In future research, it is necessary to consider the influence of regional group characteristics to confirm the model's explanatory power in different cultural contexts. Secondly, this study utilizes self-reported data. This methodology may introduce biases due to participants' potential memory distortions when recalling past experiences or behaviors, which could compromise the accuracy of the data. Consequently, future research should explore the incorporation of more objective and comprehensive data collection techniques (such as behavioral tracking, experimental observations, or multi-source validation) to either complement or substitute for self-reported data. This would significantly enhance the reliability and robustness of the research findings. Thirdly, this study only examined the direct and mediating role of HGI and TTB in scenic spots, but lacks the exploration of boundary conditions, such as individual characteristics or preferences of tourists, which may be potential moderating factors. In addition, other emotional factors, such as the quality of residents' interaction with visitors, trust, and perceived social and cultural distance, may also act as potential moderators. These topics should therefore be further explored in future work. Finally, this study selected two distinct types of case sites for investigation, with a primary aim to validate the robustness of the research findings through rigorous testing. However, it is acknowledged that different individuals frequently exhibit varying preferences or heightened sensitivity to particular factors [98]. The present study overlooked the heterogeneity of residents in its analysis. Consequently, future research should prioritize examining the heterogeneity of residents to gain a deeper understanding and more nuanced analysis of the variations in tourism choices and behaviors across different resident groups.

## Conclusion

In conclusion, despite the necessity for further replication and extension, this study has explored the impact of HGI on TTB for tourist destinations through the lenses of PA and SWB. Two empirical findings indicate that HGI can foster tourists' emotional connection to tourist destinations and enhance their overall well-being, while simultaneously increasing inclusivity towards tourist attractions. This research suggests that PA and SWB serve as potential mechanisms linking HGI to TTB. Moreover, tourists' PA and SWB function as a sequential mediator in the relationship between HGI and TTB. The mediator and multiple mediator analyses support the plausibility of the following pathway: [1] HGI→PA→TTB; [2] HGI→SWB→TTB; and [3] HGI→PA→SWB→TTB.

## Author contributions

**Conceptualization:** Yajun JIANG, Longfang HUANG, Huiling ZHOU.

**Data curation:** Yajun JIANG, Longfang HUANG.

**Formal analysis:** Longfang HUANG.

**Funding acquisition:** Huiling ZHOU.

**Investigation:** Longfang HUANG.

**Methodology:** Longfang HUANG.

**Project administration:** Huiling ZHOU.

**Software:** Yajun JIANG, Longfang HUANG, Ke WU.

**Supervision:** Huiling ZHOU.

**Validation:** Yajun JIANG, Huiling ZHOU.

**Visualization:** Yajun JIANG, Ke WU.

**Writing – original draft:** Longfang HUANG.

**Writing – review & editing:** Yajun JIANG, Longfang HUANG, Ke WU.

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
