## [Decision Letter · Decision Letter 0]

20 Aug 2024

PONE-D-24-32703Does host-guest interaction promote tolerance behavior? The mediating role of place attachment and subjective well-beingPLOS ONE

Dear Dr. Wu,

Thank you for submitting your manuscript to PLOS ONE. After careful consideration, we feel that it has merit but does not fully meet PLOS ONE’s publication criteria as it currently stands. Therefore, we invite you to submit a revised version of the manuscript that addresses the points raised during the review process.

We look forward to receiving your revised manuscript.

Kind regards,

Kun Sang, PhD

Academic Editor

PLOS ONE

Journal Requirements:

"This research was supported by The National Natural Science Foundation of China project “Research on the Impact of Social Network of Ethnic Village Farmers on Tourism Livelihood Behavior” (72064006). Institute of Guangxi Tourism Industry Scientiffc Research Fund: Research on the Inffuencing Factors of Cultural Capital of Ethnic Village Farmers on Community Citizen Behavior (LYCY2023-29);Evaluation of Livelihood Resilience of Ethnic Tourism Villages and Its Driving Mechanism(LYCY2023-06). "

Additional Editor Comments:

The academic editor agrees with the comments by the reviewers. The paper has some potential to be revised for a better version. Please refer to the comments as below.

Reviewers' comments:

Reviewer's Responses to Questions

**Comments to the Author**

1. Is the manuscript technically sound, and do the data support the conclusions?

Reviewer #1: No

Reviewer #2: Yes

2. Has the statistical analysis been performed appropriately and rigorously? 

Reviewer #1: No

Reviewer #2: Yes

3. Have the authors made all data underlying the findings in their manuscript fully available?

Reviewer #1: No

Reviewer #2: Yes

4. Is the manuscript presented in an intelligible fashion and written in standard English?

Reviewer #1: No

Reviewer #2: No

5. Review Comments to the Author

Reviewer #1: I regret to inform you that I cannot recommend this manuscript for publication. The overall writing quality does not meet the standards expected for academic work, which affects the clarity and professionalism of the paper. Additionally, there is a clear disconnect between the introduction and the literature review, with the latter failing to adequately explain the relationships between the variables under investigation. The empirical analysis is not only poorly connected to the theoretical framework but is also overly simplistic, relying on basic validation without deeper exploration. Consequently, the discussion that follows is superficial and lacks substantive insights. Due to these significant shortcomings, I must recommend rejection of this manuscript.

Reviewer #2: It is an honor to review this manuscript. This paper explores the impact of host-guest interaction on tourists' tolerance behavior and further investigates the underlying mechanisms. Overall, the structure of this paper is reasonable, and the research design is relatively satisfactory. However, there are still some issues in writing, presentation of some results, and data analysis, and I suggest that the author team work hard to improve and modify them. Specific comments are as follows:

1.First of all, it is recommended that the entire text undergo necessary language polishing. Currently, some grammar and expressions are not very academic. For example, the names of specific case locations should be capitalized, and it seems that the authors did not follow this rule in the abstract.

2.The logic of the introduction is not smooth. It is inappropriate for the author to start with tourist civic behavior as the background. It is recommended to start directly from the academic background and theoretical background of host-guest interaction or tourist tolerance.

3.The last paragraph of the introduction is not clear. The exposition of theoretical contributions is superficial. I suggest that the author reorganize the theoretical gaps and explain how they are filled.

4.Regarding the theoretical part, the SOR theory is a very familiar theory. I suggest removing the introduction of the SOR theory and placing the core views at the hypothesis derivation.

5.The derivation of the hypothesis is generally okay, but I suggest that the authors further refine it to make the derivation look more reasonable. Some expressions are not very professional, such as the subjective well-being theory on P6, I don't think it is a theory. Please check the derivation of the hypothesis again and update the necessary literature.

6.The analysis process of the two empirical studies is generally reasonable, but there are two major problems for the author to consider: First, neither of the two questionnaire studies has carried out the necessary common method bias analysis, and second, when considering the relationship between variables, the interference of irrelevant variables is not considered.

7.Regarding why these two places were chosen as case locations, I think the author should have an explanation. In addition, I think that these two studies can actually be merged into one study. You can merge them to try to draw universal conclusions, or you can use MGA to do the multi-group analysis effect comparison. Currently, study 2 is completely repeating study 1, which looks very redundant.

8.The part of theoretical contribution needs to be further deepened, and it seems that the research value of this paper is not highlighted. It is suggested that the author have a deeper dialogue with the existing literature, inform the readers what the gap is, and how your research fills it.

Good luck to the authors!

6. PLOS authors have the option to publish the peer review history of their article (what does this mean? ). If published, this will include your full peer review and any attached files.

**Do you want your identity to be public for this peer review?** For information about this choice, including consent withdrawal, please see our Privacy Policy .

Reviewer #1: No

Reviewer #2: No

---

## [Author Response · Author response to Decision Letter 1]

4 Oct 2024

Response Letter

Dear Editors and Reviewers:

Firstly, I would like to extend my sincere gratitude to you and the reviewers for the invaluable feedback and suggestions provided on my previous submission. Your comments have been instrumental not only in advancing the improvement of my research but also in significantly enhancing the overall quality of the manuscript.

Following a thorough revision process, I have comprehensively addressed each suggestion put forth by both the editorial office and reviewers. I am confident that these enhancements have substantially improved the content, logic, and clarity of the manuscript, aligning it more closely with your publication standards.

Should there be any additional areas for refinement, I will greatly appreciate your insights and strive diligently to further enhance the manuscript. Once again, I express my profound appreciation for your dedicated efforts and professional guidance. I eagerly await your positive response!

Yours sincerely,

Ke Wu

The Editors

We sincerely appreciate the time you dedicated to reviewing our manuscript and for offering invaluable feedback and suggestions regarding our research. We would like to extend our heartfelt gratitude to you.

In response to the points you've raised, we will address each one individually:

Question1: Please ensure that your manuscript meets PLOS ONE's style requirements, including those for file naming. The PLOS ONE style templates can be foundathttps://journals.plos.org/plosone/s/file?id=wjVg/PLOSOne_formatting_sample_main_body.pdfandhttps://journals.plos.org/plosone/s/file?id=ba62/PLOSOne_formatting_sample_title_authors_affiliations.pdf

Response: I sincerely appreciate your reminder regarding the requirements for PLOS ONE manuscripts, particularly concerning the file naming conventions. I have meticulously reviewed the submission guidelines for PLOS ONE and have diligently checked and adjusted my manuscript to ensure that its content, format, and file naming adhere strictly to the standards set forth by your journal.

Question2: Thank you for stating in your Funding Statement: "This research was supported by The National Natural Science Foundation of China project “Research on the Impact of Social Network of Ethnic Village Farmers on Tourism Livelihood Behavior” (72064006). Institute of Guangxi Tourism Industry Scientiffc Research Fund: Research on the Inffuencing Factors of Cultural Capital of Ethnic Village Farmers on Community Citizen Behavior (LYCY2023-29);Evaluation of Livelihood Resilience of Ethnic Tourism Villages and Its Driving Mechanism(LYCY2023-06). "Please provide an amended statement that declares *all* the funding or sources of support (whether external or internal to your organization) received during this study, as detailed online in our guide for authors at http://journals.plos.org/plosone/s/submit-now. Please also include the statement “There was no additional external funding received for this study.” in your updated Funding Statement. Please include your amended Funding Statement within your cover letter. We will change the online submission form on your behalf.

Response: I would like to extend my heartfelt appreciation for your attention to this study and your invaluable guidance regarding the specifics of the funding statement. In accordance with your request, I have undertaken a thorough revision of the funding statement to ensure it comprehensively addresses all sources of funding and support received throughout this study while adhering to the online author guidelines set forth by your journal. Below is the revised funding statement:

This research was supported by the National Natural Science Foundation of China [Grant No. 72064006, 72462013].

Question3: We note that your Data Availability Statement is currently as follows: All relevant data are within the manuscript and its Supporting Information files.Please confirm at this time whether or not your submission contains all raw data required to replicate the results of your study. Authors must share the “minimal data set” for their submission. PLOS defines the minimal data set to consist of the data required to replicate all study findings reported in the article, as well as related metadata and methods(https://journals.plos.org/plosone/s/data-availability#loc-minimal-data-set-definition).For example, authors should submit the following data:

- The points extracted from images for analysis.Authors do not need to submit their entire data set if only a portion of the data was used in the reported study.If your submission does not contain these data, please either upload them as Supporting Information files or deposit them to a stable, public repository and provide us with the relevant URLs, DOIs, or accession numbers. For a list of recommended repositories, please see https://journals.plos.org/plosone/s/recommended-repositories.If there are ethical or legal restrictions on sharing a de-identified data set, please explain them in detail (e.g., data contain potentially sensitive information, data are owned by a third-party organization, etc.) and who has imposed them (e.g., an ethics committee). Please also provide contact information for a data access committee, ethics committee, or other institutional body to which data requests may be sent. If data are owned by a third party, please indicate how others may request data access.

Response: We sincerely appreciate your comprehensive review and insightful feedback regarding our research. The findings we presented are derived from a subset of the data, which has been provided as supplementary information files. Kindly refer to the attached documents for further details.

Question4: PLOS requires an ORCID iD for the corresponding author in Editorial Manager on papers submitted after December 6th, 2016. Please ensure that you have an ORCID iD and that it is validated in Editorial Manager. To do this, go to ‘Update my Information’ (in the upper left-hand corner of the main menu), and click on the Fetch/Validate link next to the ORCID field. This will take you to the ORCID site and allow you to create a new iD or authenticate a pre-existing iD in Editorial Manager.

Response: I sincerely appreciate your comprehensive review and insightful feedback regarding our research. In accordance with PLOS requirements, I have confirmed that I, as the corresponding author, possess a valid ORCID iD, which has been duly verified in Editorial Manager.

Question5: Your ethics statement should only appear in the Methods section of your manuscript. If your ethics statement is written in any section besides the Methods, please delete it from any other section. 

Response: I would like to extend my heartfelt appreciation for your comprehensive review and insightful feedback on our study. I have meticulously examined the manuscript and confirmed that the ethical statement has been retained solely in the 'Methods' section. Additionally, I have eliminated all pertinent content related to the ethical statement from other sections of the manuscript to ensure both compliance and clarity.

Reviewer: 1

We would like to express our sincere gratitude to the reviewers for dedicating their valuable time amidst their busy schedules to evaluate our manuscript. The constructive suggestions provided by the reviewers significantly enhance the quality of our work. We extend our heartfelt appreciation for your efforts.

In response to all the points you have raised, we will address them individually as follows:

Question 1�I regret to inform you that I cannot recommend this manuscript for publication. The overall writing quality does not meet the standards expected for academic work, which affects the clarity and professionalism of the paper.

Response: We sincerely appreciate your comprehensive review and invaluable feedback on our study. In response to your earlier comment, 'The overall writing quality does not meet the standards expected for academic work, which affects the clarity and professionalism of the paper,' we have engaged in a thorough self-assessment and fully acknowledge your critique. Consequently, we have meticulously revised the sentence structure, grammar, and spelling throughout the manuscript to ensure both accuracy and fluency in our content. These enhancements are intended to improve the article's readability while effectively conveying our research findings.

Question 2: Additionally, there is a clear disconnect between the introduction and the literature review, with the latter failing to adequately explain the relationships between the variables under investigation.

Response: We sincerely appreciate your comprehensive review and invaluable feedback on our study. In response to the concern you raised that " there is a clear disconnect between the introduction and the literature review, with the latter failing to adequately explain the relationships between the variables under investigation", we have implemented targeted revisions. We have restructured the logical framework of the Introduction by first outlining the academic context of tolerance behavior, followed by a discussion of the current research landscape of tolerance behavior, an analyse of host-guest interaction dynamics, an exploration of place attachment and subjective well-being as mediating mechanisms, and a summary of the theoretical contributions of this paper". Additionally, we have eliminated redundant content to more effectively clarify these relationships among research variables. The supplementary content is detailed below:

Tolerance, as a customer behavior, is increasingly receiving widespread attention from academia[1]. Specifically, tolerance refers to the degree of forgiveness displayed by customers when service delivery fails to meet their expectations[1]. This concept emphasizes the patient and forgiving attitude of customers in service contexts[2]. Tolerance behavior not only helps customers achieve inner balance, but also provides valuable opportunities for service providers to repair potential cracks in customer relationships caused by service deficiencies[3]. Especially in the tourism industry, tolerance behavior, as a manifestation of tourist citizenship behavior [4], is of great significance for promoting the long-term stable development of the tourism industry and enhancing competitive advantages [5, 6]. Therefore, the in-depth study of TTB by academia not only enriches customer behavior theory but also provides new perspectives and strategic ideas for enhancing tourism service quality.

As TTB has increasingly emerged as a focal point of academic inquiry, scholars have investigated the multiple factors influencing TTB from diverse perspectives. These factors can primarily be divided into two dimensions: organizational and individual factors. In terms of organizational factors, research encompasses critical elements such as rule-making[7] and hotel room pricing[8]. In terms of individual factors, it addresses customers' negative emotions[9] and their perceptions of climate change [10]. However, the above research context on TTB is mainly limited to the hotel industry[7-9], given that hotels represent relatively enclosed environments where guest interactions with staff constitute the primary service engagement. Conversely, tourist attractions are characterized by more open settings in which tourists engage with guides, area personnel, and local residents in varied ways. Consequently, findings related to tolerant behavior within hotel contexts cannot be directly extrapolated to scenarios involving tourist destinations [7-9]. Furthermore, interactions between tourists and local residents at these destinations are recognized as pivotal for fostering long-term sustainable development in tourism areas[11]. Previous studies have shown that interactive experiences within tourist attractions have a significant impact on tourists' behavior, such as stimulating their participation enthusiasm[12] and promoting word-of-mouth recommendations and the formation of sustained usage intentions[13]. Therefore, it is crucial to thoroughly investigate whether—and how—the interaction between tourists and local residents in open tourist attractions impacts TTB.

To further investigate the mechanisms underlying TTB, this study introduces two psychological variables—place attachment (PA) and subjective well-being (SWB)—as significant mediators. The PA theory posits that as individuals strengthen their emotional connections to a specific location, their perceptions and behaviors towards tourism destinations are profoundly affected (14). Empirical research has demonstrated that tourists' sense of place identity and positive emotions can enhance pro-environmental behaviors (15, 16), while feelings of awe toward tourism sites directly foster helping behaviors (17). Consequently, this study seeks to elucidate the intrinsic relationship between HGI and TTB through the mediating role of PA. Additionally, drawing from emotion regulation theory, individual SWB exerts a substantial positive influence on behavior (18). In workplace contexts, SWB significantly enhances organizational citizenship behavior (18) and also impacts tourists' decision-making processes regarding travel (19). Despite numerous valuable studies examining tourists' PA emotions and SWB within academic discourse, there remains a paucity of research integrating both constructs to uncover the fundamental drivers behind tourist behavior. Therefore, this study aims to explore in greater depth and comprehensiveness the intrinsic connection between HGI and TTB via the mediating variables of PA and SWB.

The literature contribution of this study to TTB is mainly reflected in the following aspects: Firstly, it underscores the critical role of HGI in shaping TTB within tourist destinations. By establishing HGI as a fundamental prerequisite for TTB, this research addresses whether tourists who engage more interactively are indeed more tolerant. Secondly, it elucidates the underlying mechanisms through which HGI influences TTB by incorporating PA and SWB. Lastly, this investigation is situated within the context of open tourist attractions, thereby broadening the scope of research on tolerance behavior in tourism and offering a theoretical framework for exploring analogous relationships in related fields. Consequently, this study paves new avenues for future inquiries into civic behavior among tourists. More broadly, it provides essential theoretical support and practical insights aimed at enhancing TTB in tourist destinations.

The references are as follows

1.Yi Y, Gong T. Customer value co-creation behavior: Scale development and validation. J Bus Res. 2013;66(9):1279-84.

2.Yang Y, Hu J. Self-diminishing effects of awe on consumer forgiveness in service encounters. Journal of Retailing and Consumer Services. 2021;60.

3.Hur J, Jang S. Is consumer forgiveness possible? International Journal of Contemporary Hospitality Management. 2019;31(4):1567-87.

4.Zhang H, Xu H. Impact of destination psychological ownership on residents’ “place citizenship behavior”. J Destin Mark Manage. 2019;14.

5.Torres-Moraga E, Rodriguez-Sanchez C, Sancho-Esper F. Understanding tourist citizenship behavior at the destination level. J Hosp Tour Manag. 2021;49:592-600.

6.Yao Y, Wang G, Ren L, Qiu H. Exploring tourist citizenship behavior in wellness tourism destinations: The role of recovery perception and psychological ownership. J Hosp Tour Manag. 2023;55:209-19.

7.Ma Shuang LX, Li Chunqing. Customer citizenship Behavior and misbehavior in the Context of Sharing economy: from the perspective of Social Dilemma Theory. Advances in psychological science. 2022;29(11):1920-35.

8.Alderighi M, Nava CR, Calabrese M, Christille J-M, Salvemini CB. Consumer perception of price fairness and dynamic pricing: Evidence from Booking.com. J Bus Res. 2022;145:769-83.

9.Xiong Wei HM, Huang Yuanfei. The elastic change of customer's negative emotion and tolerance in the context of hotel service failure tourism science. 2021;35:53-75.

10. Seekamp E, Jurjonas M, Bitsura-Meszaros K. Influences on coastal tourism demand and substitution behaviors from clima

---

## [Decision Letter · Decision Letter 1]

11 Nov 2024

PONE-D-24-32703R1Does host-guest interaction promote tolerance behavior? The mediating role of place attachment and subjective well-beingPLOS ONE

Dear Dr. WU,

Thank you for submitting your manuscript to PLOS ONE. After careful consideration, we feel that it has merit but does not fully meet PLOS ONE’s publication criteria as it currently stands. Therefore, we invite you to submit a revised version of the manuscript that addresses the points raised during the review process.

We look forward to receiving your revised manuscript.

Kind regards,

Kun Sang, PhD

Academic Editor

PLOS ONE

Reviewers' comments:

Reviewer's Responses to Questions

**Comments to the Author**

1. If the authors have adequately addressed your comments raised in a previous round of review and you feel that this manuscript is now acceptable for publication, you may indicate that here to bypass the “Comments to the Author” section, enter your conflict of interest statement in the “Confidential to Editor” section, and submit your "Accept" recommendation.

Reviewer #1: (No Response)

Reviewer #3: (No Response)

2. Is the manuscript technically sound, and do the data support the conclusions?

Reviewer #1: Partly

Reviewer #3: Yes

3. Has the statistical analysis been performed appropriately and rigorously? 

Reviewer #1: Yes

Reviewer #3: Yes

4. Have the authors made all data underlying the findings in their manuscript fully available?

Reviewer #1: Yes

Reviewer #3: Yes

5. Is the manuscript presented in an intelligible fashion and written in standard English?

Reviewer #1: No

Reviewer #3: Yes

6. Review Comments to the Author

**Reviewer #1:**  The study explores how host-guest interaction promotes tourist tolerance behavior (TTB) through place attachment and subjective well-being. However, there are several issues that need further discussion:

First, the introduction lacks logical coherence in presenting the theoretical background and research context. It is recommended to reorganize the introduction, starting with a clear overview of the theoretical background of tolerance behavior, followed by the impact of host-guest interaction on tolerance behavior, and finally, leading to the research question and objectives.

Second, as an empirical study, I find the logical connection between Study 1 and Study 2 quite weak. Even though the authors have explained the purpose of conducting both studies, it does not adequately support the claim of generalizability for the model. Additionally, upon reviewing the latter part of the manuscript, it appears that much of the empirical content is focused on data description without adequately explaining why. This may be related to insufficient theoretical elaboration in the earlier sections, which also results in a superficial discussion in the theoretical analysis section. I recommend further referencing standardized multi-study research designs and paradigms.

Third, I have concerns about the measurement of TTB. After reviewing the cited literature, I found that there is no direct relationship between the cited scales and TTB, which raises questions about the validity of the measurement.

Lastly, the overall writing quality, both in terms of logic and style, is relatively rough and needs significant improvement.

**Reviewer #3: ** The manuscript has been well improved in the first round of revision. Here are some more detailed suggestions:

1. It would be very helpful if the authors can provide more information about the research area. What is “Two Rivers and Four Lakes” and what is “Xingping Ancient Town”? Please provide a background introduction for non-local readers.

2. Some recent research related to roles of SWB and PA would provide helpful insights to develop the theory and hypotheses. For example, “Community resilience in city emergency: Exploring the roles of environmental perception, social justice and community attachment in subjective well-being of vulnerable residents” provides a new theory combining PA (where they recognized as CA) and SWB to community development.

3. Please provide a more detailed data collection strategy and sampling methods employed for data collection. In addition, please also provide the time and duration for the data collection.

4. It would be important to consider the heterogeneity of residents in the discussion as well. For example, different people would tend to make very different choices or be more sensitive to some specific factors. There is an recent example discussing this: “Determinants and mechanisms driving energy-saving behaviours of long-stay hotel guests: Comparison of leisure, business and extended-stay residential cases”.

5. An new conclusion section rather than “discussion and conclusion” would be more helpful, as the current version would be a little bit long to summairse the core ideas of this research.

7. PLOS authors have the option to publish the peer review history of their article (what does this mean? ). If published, this will include your full peer review and any attached files.

**Do you want your identity to be public for this peer review?** For information about this choice, including consent withdrawal, please see our Privacy Policy .

Reviewer #1: No

Reviewer #3: No

---

## [Author Response · Author response to Decision Letter 2]

18 Dec 2024

Response Letter

Dear Reviewers:

Firstly, I would like to extend my sincere gratitude to you for the invaluable feedback and suggestions provided on my previous submission. Your comments have been instrumental not only in advancing the improvement of my research but also in significantly enhancing the overall quality of the manuscript.

After a meticulous revision process, I have thoroughly addressed each suggestion put forth by the reviewers. I am confident that these enhancements have substantially improved the content, logic, and clarity of the manuscript, bringing it into closer alignment with your publication standards.

If there are any additional areas for refinement, I would greatly appreciate your insights and will endeavor to further enhance the manuscript. Once again, I express my profound appreciation for your dedicated efforts and professional guidance.

I look forward to your positive response.

Yours sincerely,

Ke Wu

Reviewer: 1

We wish to convey our sincere gratitude to the reviewers for dedicating their valuable time, despite their busy schedules, to evaluate our manuscript. The constructive suggestions provided by the reviewer have significantly enhanced the quality of our work. We extend our heartfelt appreciation for your efforts.

In response to the points you have raised, we will address each one individually as follows:

Question 1�First, the introduction lacks logical coherence in presenting the theoretical background and research context. It is recommended to reorganize the introduction, starting with a clear overview of the theoretical background of tolerance behavior, followed by the impact of host-guest interaction on tolerance behavior, and finally, leading to the research question and objectives.

Response: We wish to convey our deepest appreciation to the esteemed expert for the invaluable feedback provided on our introduction section. The initial draft suffered from a lack of logical coherence when presenting the theoretical and research background. In response to the expert's recommendations, we have thoroughly restructured the introduction. Firstly, we now present a concise yet comprehensive overview of the theoretical underpinnings of tolerance behavior, encompassing its definition, significance, and the current state of research, thereby offering readers a thorough and nuanced understanding. Secondly, we delve into the influence of host-guest interactions on tolerance behavior, examining the inherent connections and underlying mechanisms, which further solidify and deepen the theoretical framework of our paper. Lastly, we articulate the research problem and objectives, delineating the research direction and focal points of the study, thus providing a clear and structured research framework for our audience. Once again, we extend our heartfelt gratitude for the meticulous guidance and insightful suggestions offered by the expert.

The modifications are outlined as follows:

In recent years, the swift advancement of an open economy and the dissemination of civilized tourism principles have prompted a growing number of tourists to willingly engage in pro-social behaviors that benefit tourism enterprises or destinations, referred to as tourist civic behaviors (1). These behaviors are characterized by their spontaneous and voluntary nature, generating value for stakeholders without being obligatory(2, 3). They encompass activities such as providing positive word-of-mouth recommendations (4), offering assistance, giving feedback(5), and demonstrating tolerance(2). Within the value co-creation framework between tourists and service providers (2), tourists' tolerance behavior assumes a pivotal role in fostering the long-term sustainable development of the tourism sector and enhancing its competitive edge (5, 6). Consequently, tolerance behavior has garnered increasing scholarly attention(2), as it not only aids in balancing the often uneven relationship between customers and service providers(7), but also actively contributes to the enhancement of tourists' travel experiences and overall tourism quality.

However, existing research on tourist tolerance behavior (TTB) predominantly focuses on the hospitality sector(8, 9, 10, 11). It is important to highlight that, within the relatively confined environment of hotels, customer interactions primarily occur between guests and hotel personnel. Conversely, tourist destinations present a more open setting where visitors engage not only with destination staff, such as tour guides and park attendants, but also regularly with local inhabitants through unstructured encounters. Consequently, the direct application of tolerance behavior influence mechanisms from the hotel context to tourist destination scenarios is constrained by these differences (8, 9, 10).

Next, we delve into the relationship between tourists and tourist destinations, with particular emphasis on the host-guest interaction (HGI), which is widely recognized as a pivotal factor in the sustainable development of tourist destinations(12). The HGI, as a distinctive tourism attraction element, significantly influences tourists' perceptions, attitudes, and behaviors (13). In tourism resources of universal value and substantial significance, such as natural and cultural heritage sites, tourists are inclined to place greater importance on the quality of host-guest interaction (14). Studies indicate that when tourists form a "warm partnership" with local residents, this relationship effectively enhances their engagement(15) and fosters the creation of positive word-of-mouth and sustained visitation intentions(4). Consequently, in both open natural heritage and cultural heritage tourist destinations, thoroughly investigating how HGI impacts TTB not only contributes to the enrichment of customer behavior theory but also offers novel perspectives and strategies for enhancing tourism service quality.

Question 2: Second, as an empirical study, I find the logical connection between Study 1 and Study 2 quite weak. Even though the authors have explained the purpose of conducting both studies, it does not adequately support the claim of generalizability for the model. Additionally, upon reviewing the latter part of the manuscript, it appears that much of the empirical content is focused on data description without adequately explaining why. This may be related to insufficient theoretical elaboration in the earlier sections, which also results in a superficial discussion in the theoretical analysis section. I recommend further referencing standardized multi-study research designs and paradigms.

Response: We wish to convey our deepest appreciation to the esteemed expert for the meticulous review of our research and the invaluable feedback provided. Concerning the weak logical connection between Study 1 and Study 2 that you have highlighted, we hold this matter in high regard and have implemented appropriate measures.

First, in this paper, Study 1 is designed to examine whether interpersonal interactions at natural heritage open-access sites influence tourists' tolerance behavior. Meanwhile, Study 2 seeks to validate the robustness of Study 1’s findings by replicating the investigation across various types of cultural heritage open-access sites, thereby enhancing the external validity of the research.

Additionally, the approach of conducting studies in diverse case sites to confirm the robustness of conclusions has been adopted by prior researchers. For instance, Su et al. executed three surveys to explore the connection between visitor service quality and subjective well-being, as well as to assess the consistency of the proposed model across different destinations.

He, Xuehuan, Lujun Su, and Scott R. Swanson. "The service quality to subjective well-being of Chinese tourists connection: A model with replications." Current Issues in Tourism.2020, 23(16): 2076-2092.

Furthermore, we have refined and adjusted the descriptions of Study 1 and Study 2 to ensure the overall coherence and persuasiveness of the research.

The modifications are outlined as follows:

Study Site section: Finally, gathering samples from two distinct categories of tourist attractions and performing independent data analyses contributes to enhancing the validity and robustness of our research outcomes.

Measurement model part of Study 1: By analysing the data with Amos software, the measurement model of study 1 was well fitted (see Table 3).Cronbach's alpha values ranged from 0.830 to 0.938, all greater than 0.7, meeting the criteria(73). Factor loadings for all items (see Table 4) were greater than 0.5, which is statistically significant (p < 0.001) and meets the criterion(73). The composite reliability (CR) values ranged from 0.827 to 0.938, with each variable's CR exceeding the minimum threshold of 0.7 as recommended by Hair et al.(74). Additionally, the average variance extracted (AVE) for all constructs ranged from 0.549 to 0.791, satisfying the established criteria(73). Therefore, in terms of the overall data, the samples in Study 1 all have high reliability, validity and internal consistency.

Data collection part of Study 2: To examine the proposed model’s stability across destinations.

Measurement model part of Study 2: The measurement model of Study 2 was well fitted (see Table 3), Cronbach's alpha values ranging from 0.864 to 0.924, all exceeding the threshold of 0.7, thus satisfying the reliability criteria. Factor loadings for all items, as presented in Table 6, were greater than 0.5 and achieved statistical significance (p < 0.001). The composite reliability (CR) of each variable ranged from 0.866 to 0.918, surpassing the 0.7 benchmark, while the average variance extracted (AVE) for all constructs ranged from 0.618 to 0.807, well above the minimum requirement of 0.5(73). Therefore, from perspective of the overall data, the samples in Study 2 all have high reliability, validity and internal consistency.

Concerning the critique that the empirical content was overly focused on data description and lacked in-depth analysis, we have undertaken a thorough revision of the discussion section. By integrating real-world case studies and theoretical frameworks, we now offer a more profound and nuanced interpretation and examination of our research findings, aiming to provide readers with richer and more insightful analytical perspectives.

The modifications are outlined as follows:

Discussion

This study selected tourists from two distinct tourist destinations as research subjects to explore the intrinsic relationships between HGI, PA, SWB, and TTB through the construction of a conceptual model. By systematically collecting and analyzing feedback from tourists at these destinations, this study has not only enriched the body of knowledge on tolerance behavior but also expanded the existing research scope.

Firstly, we have confirmed that HGI exerts a significant positive influence on TTB. The HGI is considered a key driver in promoting the long-term sustainable development of tourist destinations(12). Prior research has predominantly examined HGI as a prerequisite or moderating variable influencing residents' behaviors (24), with a focus on its effects on tourists' green behaviors(21) and environmentally responsible actions (22). Nevertheless, the mechanisms through which HGI impacts TTB are more intricate and multifaceted, and existing research remains inadequate. Tolerant behavior not only facilitates tourists' internal harmony but also offers valuable opportunities for service providers to rectify damaged customer relationships resulting from service failures (76).

Secondly, our research indicates that PA serves as a mediating factor between HGI TTB. PA signifies the evolving emotional connection between tourists and tourist destinations (31), while HGI enhances this attachment. This finding aligns with prior studies demonstrating that HGI significantly influence tourists' cognitive and emotional responses(33). Furthermore, the strengthening of the human-place emotional bond positively impacts TTB, corroborating existing research that tourists' emotional attachment and sense of identity shape their attitudes toward tourist destinations, thereby influencing their behaviors and decisions(39). We propose that in the context of open tourist destinations, where tourists have easy access to communication with local residents, this interaction facilitates the reinforcement of the emotional bond between tourists and destinations, leading to greater willingness to tolerate imperfections or negative aspects of the tourist experience.

Thirdly, we have confirmed that SWB acts as a partial mediator between HGI and TTB. SWB represents individuals' cognitive assessments of their own life satisfaction (42). Prior research predominantly concentrated on the influence of HGI on residents' well-being(50, 51), whereas this study broadens the scope to include tourists at tourism destinations, offering a novel perspective on the positive effects of HGI on tourists. Our findings suggest that within the tourism context, HGI can enhance tourists' SWB, and when tourists experience elevated levels of well-being at their destinations, they are more inclined to exhibit understanding and tolerance. This aligns with the theory of emotion regulation, which posits that an individual's emotional state substantially impacts their behavior (54). Specifically, the communication and interaction between tourists and local residents at tourism destinations not only influence the quality of tourists' travel experiences but also profoundly affect their SWB. Tourists with higher SWB levels are more likely to adopt a positive attitude towards tourism-related unfavorable situations.

To address the issue of insufficient theoretical elaboration, which has led to a superficial theoretical analysis, we have implemented corresponding enhancements. In the theoretical analysis section, we have expanded the depth of our analysis of research findings and the extraction of theoretical contributions, with the aim of solidifying and deepening the theoretical foundation of the study. Additionally, we have proactively adopted standardized multi-research designs and paradigms to ensure the full scientific rigor and robustness of the study.

The modifications are outlined as follows:

Theoretical implications

This study makes several contributions. Firstly, by investigating the relationships among tourists' HGI, PA, SWB, and TTB, it not only broadens the scope of TTB research and its antecedents but also provides valuable insights for destination managers aiming to enhance destination development. While previous literature on tolerance behavior has predominantly centered on hotel environments(8, 9, 10), it has not extensively explored the causal relationship between open-space tourists' HGI and TTB through multiple case studies. It is important to note that open-space tourism destinations differ significantly from closed-space hotels, where customer expectations are primarily focused on highly personalized services, and their tolerance is mainly directed towards minor service errors or shortcomings. In contrast, open-space tourists seek to experience natural beauty, cultural heritage, or entertainment activities, and their tolerance is more often related to inconveniences encountered during their tourism experiences. Considering the positive influence of tourists on destination development (5, 6), this study innovatively extends the application of TTB research by validating the impact of open-space tourists' HGI on TTB.

Secondly, this study selected two distinct types of tourist destinations for data collection: the Two Rivers and Four Lakes, which are natural heritage sites, and Xingping Ancient Town, a cultural heritage site. This selection not only encompasses the diversity of tourist destinations but also offers a valuable empirical foundation for investigating the similarities and differences in visitor behavior patterns across different types of tourist destinations. The data analysis from these two case study sites underscores the broad applicability and robustness of the research findings. It enhances the credibility of the study and provides a solid foundation for future research to prom

---

## [Decision Letter · Decision Letter 2]

24 Jan 2025

PONE-D-24-32703R2Does host-guest interaction promote tolerance behavior? The mediating role of place attachment and subjective well-beingPLOS ONE

Dear Dr. WU,

Thank you for submitting your manuscript to PLOS ONE. After careful consideration, we feel that it has merit but does not fully meet PLOS ONE’s publication criteria as it currently stands. Therefore, we invite you to submit a revised version of the manuscript that addresses the points raised during the review process.

We look forward to receiving your revised manuscript.

Kind regards,

Kun Sang, PhD

Academic Editor

PLOS ONE

Additional Editor Comments:

We agree with the comments by reviewers, especially reviewer 1. The paper still need more modifications accordingly.

Reviewers' comments:

Reviewer's Responses to Questions

**Comments to the Author**

1. If the authors have adequately addressed your comments raised in a previous round of review and you feel that this manuscript is now acceptable for publication, you may indicate that here to bypass the “Comments to the Author” section, enter your conflict of interest statement in the “Confidential to Editor” section, and submit your "Accept" recommendation.

Reviewer #1: (No Response)

Reviewer #3: All comments have been addressed

2. Is the manuscript technically sound, and do the data support the conclusions?

Reviewer #1: Yes

Reviewer #3: (No Response)

3. Has the statistical analysis been performed appropriately and rigorously? 

Reviewer #1: Yes

Reviewer #3: Yes

4. Have the authors made all data underlying the findings in their manuscript fully available?

Reviewer #1: Yes

Reviewer #3: Yes

5. Is the manuscript presented in an intelligible fashion and written in standard English?

Reviewer #1: No

Reviewer #3: Yes

6. Review Comments to the Author

Reviewer #1: Despite the revisions, I remain concerned about several aspects of the manuscript that impact its suitability for publication. First, the theoretical model requires further development. The application of the SOR model to this context, particularly the role of HGI as the stimulus, needs to be more clearly articulated and justified. The choice of PA and SWB as mediators, and their proposed chain relationship, also needs stronger theoretical support. Second, the measurement of HGI relies solely on self-reported data, which is potentially problematic. Exploring more objective or nuanced measures of HGI would strengthen the study's findings. Third, the rationale for conducting two separate studies is not entirely convincing. A more integrated approach, or a clearer articulation of the unique contribution of each study, would be beneficial. I believe addressing these core issues is crucial for improving the overall quality and impact of the manuscript.

Reviewer #3: (No Response)

7. PLOS authors have the option to publish the peer review history of their article (what does this mean? ). If published, this will include your full peer review and any attached files.

**Do you want your identity to be public for this peer review?** For information about this choice, including consent withdrawal, please see our Privacy Policy .

Reviewer #1: No

Reviewer #3: No

---

## [Author Response · Author response to Decision Letter 3]

1 Mar 2025

Response Letter

Dear Editor:

First of all, we would like to extend our sincere gratitude to the reviewer for their insightful comments and constructive suggestions on our previously submitted manuscript. Their feedback has been invaluable, not only significantly enhancing the quality of my research but also elevating the overall standard of the manuscript.

Following a thorough revision process, we have meticulously addressed each of the reviewers' recommendations, resulting in comprehensive improvements to the manuscript. These enhancements have substantially strengthened the content, logic, and clarity of expression, bringing the manuscript into closer alignment with the publication standards of your esteemed journal. Should there be any further areas requiring refinement, we would greatly appreciate your guidance and will endeavor to make additional improvements accordingly.

Once again, we express our profound appreciation for your diligent work and professional expertise. We look forward to your valuable response..

Yours sincerely,

Ke Wu

Reviewer: 1

We would like to extend our sincere appreciation to you for dedicating your valuable time to evaluate our manuscript. The constructive feedback provided has significantly enhanced the quality of our submission. Herein, we wish to convey our deepest gratitude.

With respect to all the points raised, we will address each one sequentially.

Question 1�First, the theoretical model requires further development. The application of the SOR model to this context, particularly the role of HGI as the stimulus, needs to be more clearly articulated and justified. The choice of PA and SWB as mediators, and their proposed chain relationship, also needs stronger theoretical support.

Response: We are most grateful for your thorough review of our research and for offering valuable feedback. Specifically, concerning the suggestion that the theoretical model requires further development, particularly in clearly explaining and substantiating the application of the SOR (Stimulus-Organism-Response) model and the role of host-guest interaction (HGI) as a stimulus factor, we have made the following revisions. In the "Theoretical Basis and Hypothesis" section, we have expanded on the foundational principles of SOR theory and its applications across various domains. We have meticulously analyzed the mechanism by which interaction functions as a stimulus factor within the SOR model. Through an extensive review of relevant literature, we have identified that interaction significantly influences an individual's psychological state (Organism), thereby guiding specific behavioral responses (Response). This insight provides a robust theoretical foundation for our subsequent research and enables us to construct and validate the applicability of the SOR model in the context of host-guest interaction more rigorously.

SOR Theory

In 1974, Mehrabian and Russell expanded upon the S-O-R theory within the framework of environmental psychology.They posited that various elements of the external environment serve as stimulus factors(S), which influence an individual's cognitive, emotional, and physiological states(O), ultimately shaping the individual's attitudes and behavioral responses(R)(16). The organism, representing the internal emotional and cognitive condition of tourists(17), acts as a mediator between external environmental stimuli and tourists' behavioral responses(16).

The validity of the SOR theory has been extensively empirically tested across various domains. Existing research demonstrates that emotional interaction, characterized by familiarity and intimacy as environmental stimuli, plays a pivotal role in influencing users' purchase intention processes(18). Interactions among users positively impact user perception, facilitating a deeper understanding of the product and enhancing user behavior(19). In marketing, leveraging social media can augment customers' SWB, thereby fostering increased brand loyalty(20). Within the tourism sector, positive HGI, destination image, trust, and attitudes positively promote intentions to support tourism(21). Furthermore, natural empathy and perceived environmental responsibility mediate the relationship between online interactions and consumers' intentions toward low-carbon tourism behaviors(22).

Nonetheless, there remains a paucity of research examining the influence of HGI on TTB. Consequently, this study employs the SOR theory as its theoretical framework. Specifically, HGI serves as the stimulus factor, while place attachment (PA) and SWB function as organism factors reflecting tourists' internal emotional states. TTB is considered the behavioral response. This paper constructs a structural equation model to investigate not only the impact of HGI on TTB but also the mediating role of PA and SWB in this relationship.

Secondly, concerning the issue you raised regarding the selection of PA and SWB as mediator variables and the need for stronger theoretical support for their proposed chained relationship, we have provided specific enhancements and in-depth elaborations in the "Theoretical Basis and Hypothesis" section. Specifically, we have thoroughly introduced the core concepts of attachment theory and meticulously analyzed the relationships between PA and SWB, PA and behavior, and SWB and behavior within this theoretical framework. We have further elucidated how PA and SWB influence behavior through a sequential mediation process. These additions aim to provide readers with a more comprehensive and profound understanding of the theoretical underpinnings for our variable selection and the interaction mechanisms among these variables. Additionally, we have restructured and integrated the relevant literature and logical arguments in "The mediating role of PA" and "The chain mediating effect of PA and SWB" sections based on attachment theory to ensure that our research is theoretically rigorous and complete. We trust that these modifications will more accurately address your comments and enhance the theoretical depth and persuasiveness of our study. A brief summary of the supplementary content follows:

Attachment Theory

Attachment theory, which stems from the mother-infant bond theory, is a pivotal framework in the study of human relationships and provides a comprehensive structure for understanding the development of emotional bonds(23, 24). Attachment refers to the inherent human tendency to form emotional connections with specific entities, and this inclination can significantly influence an individual's judgment. Scholars generally agree that the attachment bond between individuals and places can foster a sense of security and well-being(25). Hong's research demonstrates a positive correlation between an individual's value attachment and SWB(26). Ujang further highlights that tourists' PA can enhance the generation of well-being(27). DiWu et al. found that the impact of tourists' PA on SWB varies across different dimensions(28).

Emotion, as a fundamental component of the human-environment relationship (29), occupies a pivotal role in attachment theory. This theory suggests that subjective emotional states significantly influence human behavioral choices(30). For example, Wang et al. utilized attachment theory to conduct an in-depth analysis of the social attachment factors underlying community members' purchasing behaviors in business activities(31). Shallcross et al. examined the potential link between attachment orientation and the propensity to share positive news within the framework of attachment theory(32). Cohn posited that well-being serves as an external manifestation of emotional states and can directly impact behavioral outcomes(33). Individuals experiencing positive emotions tend to exhibit higher levels of well-being, which in turn positively reinforces behavioral outcomes(33). Furthermore, research indicates that SWB mediates the relationship between PA and pro-environmental behavior(34). Consequently, this paper hypothesizes that PA not only influences SWB but also that both PA and SWB may function as mediating variables in the process by which HGI affects TTB.

The mediating role of PA

Attachment theory posits that PA, defined as a deep emotional bond between individuals and their environments, is a critical psychological construct that substantially influences tourists' behaviors and decision-making processes(54). Wang et al. utilized attachment theory to elucidate the motivations behind purchasing behaviors among community members in commercial contexts and evaluated the influence of social attachment on such behaviors(31); Shallcross et al. explored the correlation between attachment orientation and the propensity to share positive news within the framework of attachment theory(32). Moreover, prior research has demonstrated that PA exerts a positive and significant effect on tourists' environmental responsibility behaviors(55), behavioral intentions(56), and revisit intentions(57). Building on this body of literature, this study hypothesizes that PA can positively impact TTB.

The chain mediating effect of PA and SWB

Attachment theory is not only pertinent to interpersonal relationships(78), but it also serves as a framework for elucidating the emotional connection between individuals and their environments(79, 80). Empirical evidence demonstrates that tourists can develop a distinctive emotional attachment to their travel destinations, which significantly influences their SWB(27, 28). More specifically, the emotional bond formed between an individual and a particular region can provide psychological security and a sense of social belonging(30), thereby effectively enhancing SWB(81, 82). Furthermore, research indicates that PA is linked to both physical and mental health(83, 84) and can foster positive emotional and behavioral responses(85, 86). Based on these findings, this study proposes the following hypothesis:

H8).PA has a significant positive effect on SWB.

In prior research, PA and SWB have been extensively validated as critical mediating factors. Specifically, PA mediates the relationship between local social identity and well-being(87), while community attachment serves as a mediator between community environmental perception and residents' SWB(88). Furthermore, Lin highlighted that SWB functions as an intermediary between PA and pro-environmental behavior(34). Wang demonstrated that PA and SWB exert a sequential mediating effect on the influence of environmental cognition on the loyalty of rural homestay summer vacationers(89). Drawing on SOR theory, attachment theory, and the aforementioned hypotheses, we posit that HGI, as an external stimulus, can influence TTB by enhancing their PA and SWB. Based on these theoretical frameworks and empirical findings, this study proposes the following hypothesis:

H9).PA and SWB play a chain-mediated role between HGI and TTB.

Question 2: Second, the measurement of HGI relies solely on self-reported data, which is potentially problematic. Exploring more objective or nuanced measures of HGI would strengthen the study's findings.

Response: We are deeply appreciative of your meticulous review of our research and the invaluable suggestions provided. With regard to your concern that the measurement of HGI is solely reliant on self-reported data, we offer the following response:

The methodology for measuring HGI is grounded in established approaches from prior studies (e.g., Stylidis et al., 2020; Qu et al., 2024), which provide a robust theoretical foundation and methodological support for our investigation. To address potential common method bias, we have included additional test results in the manuscript, specifically employing Harman's single-factor test. The findings indicate that common method bias has not substantially influenced our research outcomes. However, we acknowledge the inherent limitations of self-reported data and recognize this as a constraint of our current study. Consequently, in the "Limitations and future research directions" section, we have highlighted this issue and proposed exploring more objective and comprehensive data collection methods (such as behavioral data, experimental data, or multi-source data) to supplement or replace self-reported data in future research. We believe these enhancements will significantly bolster the reliability and scientific rigor of our research findings. The supplementary content is detailed below:

Secondly, this study utilizes self-reported data. This methodology may introduce biases due to participants' potential memory distortions when recalling past experiences or behaviors, which could compromise the accuracy of the data. Consequently, future research should explore the incorporation of more objective and comprehensive data collection techniques (such as behavioral tracking, experimental observations, or multi-source validation) to either complement or substitute for self-reported data. This would significantly enhance the reliability and robustness of the research findings.

Stylidis D. Exploring resident–tourist interaction and its impact on tourists’ destination image[J]. Journal of Travel Research, 2022, 61(1): 186-201.

Qu Y, Zhou Q, Cao L. How do positive host-guest interactions in tourism alter the indicators of tourists’ general attachment styles? A moderated mediation model[J]. Tourism Management, 2024, 105: 104937.

Question 3:Third, the rationale for conducting two separate studies is not entirely convincing. A more integrated approach, or a clearer articulation of the unique contribution of each study, would be beneficial.

Response: We express our sincere gratitude for your thorough review of our research and the invaluable suggestions offered.

In this study, the primary objective of conducting two independent studies is to validate the robustness of our research findings. By systematically collecting and analyzing data from both natural heritage sites and cultural heritage sites, we can comprehensively examine the relationships among variables. A literature review reveals that other scholars have similarly employed this method to verify the robustness of their conclusions. For example, Su conducted surveys of tourists in Xiamen and Yuelu Mountain to investigate the relationship between service quality and subjective well-being, providing valuable methodological support for our study. Rather than discussing the results of each study separately, we opted for a comprehensive analysis to enhance the generalizability and robustness of our conclusions. The findings from both studies indicate consistent relationships among variables across different types of scenic spots, further supporting our research outcomes. Additionally, to avoid excessive length, we chose to emphasize the commonalities and unique significance of the two studies in the integrated discussion.

We trust that our explanation will alleviate any concerns you may have and we anticipate your valuable feedback and guidance on our research with keen interest.

He X, Su L, Swanson S R. The service quality to subjective well-being of Chinese tourists connection: A model with replications[J]. Current Issues in Tourism, 2020, 23(16): 2076-2092.

For your reference, we have highlighted the revised sections in blue. We sincerely appreciate your valuable feedback and suggestions. We remain committed to continuously enhancing the quality of our research. Should you have any further recommendations for improvement, we would be most grateful to receive them and will endeavor to implement enhancements accordingly. Thank you once again for your diligent efforts and professional guidance.

---

## [Decision Letter · Decision Letter 3]

8 Apr 2025

Does host-guest interaction promote tolerance behavior? The mediating role of place attachment and subjective well-being

PONE-D-24-32703R3

Dear Dr. Wu,

We’re pleased to inform you that your manuscript has been judged scientifically suitable for publication and will be formally accepted for publication once it meets all outstanding technical requirements.

Kind regards,

Kun Sang, PhD

Academic Editor

PLOS ONE

Additional Editor Comments (optional):

Reviewers' comments:

Reviewer's Responses to Questions

**Comments to the Author**

1. If the authors have adequately addressed your comments raised in a previous round of review and you feel that this manuscript is now acceptable for publication, you may indicate that here to bypass the “Comments to the Author” section, enter your conflict of interest statement in the “Confidential to Editor” section, and submit your "Accept" recommendation.

Reviewer #1: All comments have been addressed

2. Is the manuscript technically sound, and do the data support the conclusions?

Reviewer #1: Yes

3. Has the statistical analysis been performed appropriately and rigorously? 

Reviewer #1: Yes

4. Have the authors made all data underlying the findings in their manuscript fully available?

Reviewer #1: Yes

5. Is the manuscript presented in an intelligible fashion and written in standard English?

Reviewer #1: Yes

6. Review Comments to the Author

Reviewer #1: All comments had been solved. Green light now. I don't have any more feedback.

7. PLOS authors have the option to publish the peer review history of their article (what does this mean? ). If published, this will include your full peer review and any attached files.

**Do you want your identity to be public for this peer review?** For information about this choice, including consent withdrawal, please see our Privacy Policy .

Reviewer #1: No

---

## [Editor Report · Acceptance letter]

PONE-D-24-32703R3

PLOS ONE

Dear Dr. WU,

I'm pleased to inform you that your manuscript has been deemed suitable for publication in PLOS ONE. Congratulations! Your manuscript is now being handed over to our production team.

Kind regards,

on behalf of

Dr. Kun Sang

Academic Editor

PLOS ONE